# Greedy Algorithm for Structured Bandits: A Sharp Characterization of Asymptotic Success / Failure

**Aleksandrs Slivkins**
Microsoft Research, New York City
slivkins@microsoft.com

**Yunzong Xu**
University of Illinois Urbana-Champaign
xyz@illinois.edu

**Shiliang Zuo**
University of Illinois Urbana-Champaign
szuo3@illinois.edu

## Abstract

We study the greedy (exploitation-only) algorithm in bandit problems with a known reward structure. We allow arbitrary finite reward structures, while prior work focused on a few specific ones. We fully characterize when the greedy algorithm asymptotically succeeds or fails, in the sense of sublinear vs. linear regret as a function of time. Our characterization identifies a partial identifiability property of the problem instance as the necessary and sufficient condition for the asymptotic success. Notably, once this property holds, the problem becomes easy—*any* algorithm will succeed (in the same sense as above), provided it satisfies a mild non-degeneracy condition. Our characterization extends to contextual bandits and interactive decision-making with arbitrary feedback. Examples demonstrating broad applicability and extensions to infinite reward structures are provided.

## 1 Introduction

Online learning algorithms often face uncertainty about the counterfactual outcomes of their actions. To navigate this uncertainty, they balance two competing objectives: *exploration*, making potentially suboptimal decisions to acquire information, and *exploitation*, leveraging known information to maximize rewards. This trade-off is central to the study of *multi-armed bandits* [Slivkins, 2019, Lattimore and Szepesvári, 2020], a foundational framework in sequential decision-making.

While exploration is central to bandit research, it presents significant challenges in practice, esp. when an algorithm interacts with human users. First, exploration can be wasteful and risky for the current user, imposing a burden that may be considered unfair since its benefits primarily accrue to future users. Second, exploration adds complexity to algorithm design,and its adoption in large-scale applications requires substantial buy-in and engineering support compared to a system that only exploits [Agarwal et al., 2016, 2017]. Third, exploration may be incompatible with users' incentives when actions are controlled by the users. *E.g.,* an online platform cannot *force* users to try and review new products; instead, users gravitate toward well-reviewed or familiar options [Kremer et al., 2014].[1]

A natural alternative is the *greedy algorithm* (Greedy), which exploits known information at every step without any intentional exploration. This approach sidesteps the aforementioned challenges and often better aligns with user incentives. In particular, it models the natural dynamics in an online

---

[1]. Enforcing exploration in such settings is very challenging (Kremer et al. [2014] and follow-up work, see Slivkins [2023] for an overview). While exploration *can* be made incentive-compatible, doing so involves considerable performance and/or monetary costs and additional complexity. More importantly, it hinges upon substantial (even if standard) assumptions from economic theory.

39th Conference on Neural Information Processing Systems (NeurIPS 2025).

platform where each user acts in self-interest, making decisions based on full observations of previous users' actions and outcomes, *e.g.,* purchases and product reviews [Acemoglu et al., 2022].

Despite its simplicity and practical appeal, `Greedy` is widely believed to perform poorly. This belief is deeply ingrained in the bandit literature, which overwhelmingly focuses on exploration as a necessary ingredient for minimizing regret. A key motivation for this focus comes from well-known failure cases in *unstructured $K$-armed bandits*. A classic example is as follows: "Suppose the reward of each arm follows an independent Bernoulli distribution with a fixed mean, and `Greedy` is initialized with a single sample per arm. If the best arm initially returns a $0$ while another arm returns a $1$, `Greedy` permanently excludes the best arm."

However, beyond such examples, the broader picture remains murky, especially for the widely-studied *structured bandits* – bandit problems with a known reward structure (e.g., linearity, Lipschitzness, convexity) – where observing some actions provides useful information about others. Formally, a reward structure restricts the possible *reward functions* that map arms to their mean rewards. Reward structures reduce the need for explicit exploration, making the bandit problem more tractable. For *some* of them, `Greedy` in fact succeeds, *e.g.,* two-armed bandits with expected rewards that sum up to a known value. The literature provides a few examples of failure for some specific (one-dimensional, linear) reward structures, and a few non-trivial examples of success (*e.g.,* for linear contextual bandits); see more on this in Related Work. Likewise, large-scale experiments yield mixed results: some settings confirm the need for exploration, but others indicate that `Greedy` performs well [Bietti et al., 2021]. This contrast raises a fundamental question: *When—and why—does* `Greedy` *fail or succeed?*

**Our Contributions.** We work towards the missing foundation for structured bandits: a general theory of `Greedy`. Our main result allows finite, but otherwise *arbitrary* reward structures. We provide a complete characterization of when `Greedy` asymptotically *fails* (incurs linear regret) vs when it *succeeds* (achieves sublinear regret). Our characterization applies to *every* problem instance, resolving it in the positive or negative direction, not (just) in the worst case over a particular reward structure. The negative results are of primary interest here, as they substantiate the common belief that `Greedy` performs poorly, and the positive results serve to make the characterization precise.

A key insight is identifying a new "partial identifiability" property of the problem instance, called *self-identifiability*, as a *necessary and sufficient* condition for the asymptotic success. Self-identifiability asserts that, given the reward structure, fixing the expected reward of a suboptimal arm uniquely identifies it as suboptimal. We prove that `Greedy` achieves sublinear regret under self-identifiability, and suffers from linear regret otherwise. The negative result is driven by the existence of a *decoy*: informally, an alternative reward model such that its optimal arm is suboptimal for the true model and both models coincide on this arm. We show that with some positive probability, `Greedy` gets permanently stuck on such a decoy, for an infinite time horizon. For the positive result, `Greedy` succeeds (only) because self-identifiability makes the problem instance intrinsically "easy". In fact, this success is *not* due to any particular cleverness of `Greedy`: we show that *any* reasonable algorithm (satisfying a mild non-degeneracy condition) achieves sublinear regret under self-identifiability.

Our characterization allows for essentially an arbitrary interaction protocol between the algorithm and the environment (Section 5). Specifically, we handle the model of "decision-making with structured observations" (`DMSO`, Foster et al. [2021]), which allows for arbitrary auxiliary feedback after each round. This model subsumes contextual bandits, combinatorial semi-bandits, and bandits with graph-based feedback, as well as episodic reinforcement learning. Before moving to this full generality, our presentation focuses on contextual bandits, where we obtain quantitatively stronger guarantees (Section 4), and "vanilla" bandits as a paradigmatic case for building key intuition and cleaner definitions (Section 3).

We apply our machinery to several examples, both positive and negative (Appendix D). We demonstrate that most infinite structures of interest admit meaningful finite analogs via discretization. We find that `Greedy` fails in linear bandits, Lipschitz bandits and "polynomial bandits" (with arms in $\mathbb{R}$ and polynomial expected rewards), and does so for *almost all* problem instances. For linear *contextual* bandits, `Greedy` succeeds if the context set is "sufficiently diverse", but may fail if it is "low-dimensional". For *Lipschitz* contextual bandits, `Greedy` behaves very differently, failing for almost all instances. One informal takeaway is that `Greedy` fails as a common case for most/all bandit structures of interest, whereas for *contextual* bandits it can go either way, depending on the structure. The success of `Greedy` appears to require context diversity and a parametric reward structure.

The second main result of this work concerns *infinite* (*e.g.,* continuous) reward structures (Section 6). While our earlier analysis gave a sharp "if and only if" characterization for finite reward structures, such a complete characterization is more challenging to obtain for infinite ones. To make progress, we provide a characterization parameterized by a notion of "margin" (separating instances for which the positive result applies from instances for which the negative result applies), with guarantees that deteriorate as the margin vanishes. Subject to this margin, our result handles arbitrary infinite reward structures. It applies to structured bandits with finite action sets, requiring stronger notions of self-identifiability and decoy existence (parameterized by the margin), as well as new analysis ideas.

**Discussion.** The distinction between linear and sublinear regret is a standard "first-order" notion of success vs failure in bandits. Our positive results attain logarithmic, instance-dependent regret rates, possibly with a large multiplicative constant determined by the reward structure and the instance. Our negative results establish a positive (but possibly very small) constant probability of a "failure event" where `Greedy` gets permanently stuck on a decoy, for an infinite time horizon. Optimizing these constants for an arbitrary reward structure appears difficult. However, we achieve much better constants for the partial characterization in Section 6.

The greedy algorithm is initialized with some warm-up data collected from the same problem instance (and it needs at least 1 warm-up sample to be well-defined). Our negative results require exactly one warm-up sample for each context-arm pair. All our positive results allow for an arbitrary amount of initial data. Thus, our characterization effectively defines "success" as sublinear regret for any amount of warm-up data, and "failure" as linear regret for *some* amount of warm-up data.

We assume that `Greedy` is given a *regression oracle*: a subroutine to perform (least-squares) regression given the reward structure. As in "bandits with regression oracles" (referenced below), we separate out computational issues, leveraging prior work on regression, and focus on the statistical guarantees.

**Related Work.** Bandit reward structures studied in prior work include linear and combinatorial structures [*e.g.,* Awerbuch and Kleinberg, 2008, McMahan and Blum, 2004, György et al., 2007, Cesa-Bianchi and Lugosi, 2012], convexity [*e.g.,* Kleinberg, 2004, Flaxman et al., 2005, Bubeck et al., 2017], and Lipschitzness [*e.g.,* Kleinberg, 2004, Kleinberg et al., 2008, Bubeck et al., 2011], as well as some others. Each of these is a long line of work on its own, with extensions to contextual bandits [*e.g.,* Li et al., 2010, Slivkins, 2014]. There's also some work on bandits with *arbitrary* reward structures [Amin et al., 2011, Combes et al., 2017, Jun and Zhang, 2020, Degenne et al., 2020, Parys and Golrezaei, 2024], and particularly contextual bandits with regression oracles [*e.g.,* Agarwal et al., 2012, Foster et al., 2018, Foster and Rakhlin, 2020, Simchi-Levi and Xu, 2022]. For more background, see books Slivkins [2019], Lattimore and Szepesvári [2020], Foster and Rakhlin [2023].

For `Greedy`, positive results with near-optimal regret rates focus on linear contextual bandits with diverse/smoothed contexts [Kannan et al., 2018, Bastani et al., 2021, Raghavan et al., 2023, Kim and Oh, 2024]. Both context diversity and parametric reward structure are essential. Our positive results for the same setting are incomparable: more general in terms of context diversity assumptions, but weaker in terms of the regret bounds. (This line of prior work does not contain any negative results.) `Greedy` is also known to attain $o(T)$ regret in various scenarios with a very large number of near-optimal arms [Bayati et al., 2020, Jedor et al., 2021].[2]

Negative results for `Greedy` are derived for "non-structured" $K$-armed bandits: from trivial extensions of the single-sample-per-arm example mentioned above, to an exponentially stronger characterization of failure probability [Banihashem et al., 2023], to various "near-greedy" algorithms / behaviors, both "frequentist" and "Bayesian" (same paper). Negative results for non-trivial reward structures concern dynamic pricing with linear demands [Harrison et al., 2012, den Boer and Zwart, 2014] and dynamic control in a generalized linear model [Lai and Robbins, 1982, Keskin and Zeevi, 2018]. Banihashem et al. [2023] also obtain negative results for the Bayesian version of `Greedy` in Bayesian bandits, under a certain "full support" assumption on the prior.[3]

## 2   Preliminaries: structured contextual bandits (`StructuredCB`)

We have *action set* $\mathcal{A}$ and *context set* $\mathcal{X}$. In each round $t = 1, 2, \ldots$, a context $x_t \in \mathcal{X}$ arrives, an algorithm chooses an action (*arm*) $a_t \in \mathcal{A}$, and a reward $r_t \in \mathbb{R}$ is realized. The context is drawn

---

2. *E.g.,* for Bayesian bandits with $\gg \sqrt{T}$ arms, where the arms' mean rewards are sampled uniformly.
3. Essentially, the prior covers all reward functions {arms} $\to [0, 1]$ with probability density at least $p > 0$.

independently from some fixed and known distribution over $\mathcal{X}$. [4] The reward $r_t$ is an independent draw from a unit-variance Gaussian with unknown mean $f^*(x_t, a_t) \in [0, 1]$. [5] A *reward function* is a function $f : \mathcal{X} \times \mathcal{A} \to [0, 1]$; in particular, $f^*$ is the *true* reward function. The reward structure is given by a known class $\mathcal{F}$ of reward functions which contains $f^*$; the assumption $f^* \in \mathcal{F}$ is known as *realizability*. To recap, the problem instance is a pair $(f^*, \mathcal{F})$, where $\mathcal{F}$ is known and $f^*$ is not.

We focus on *finite* reward structures, *i.e.,* assume (unless specified otherwise) that $\mathcal{X}, \mathcal{A}, \mathcal{F}$ are all finite. While this does not hold for most reward structures from prior work, one can discretize them to ensure finiteness. Indeed, when reward functions can take infinitely many values, one could round each function value to the closest point in some finite subset $S \subset [0, 1]$, *e.g.,* all integer multiples of some $\varepsilon > 0$. Likewise, one could discretize contexts, arms, or function parameters, when they are represented as points in some metric space, *e.g.,* as real-valued vectors. Or, one could define finite reward structures *directly*, with similar discretizations built-in (see Appendix D for examples).

We are interested in expected regret $\mathbb{E}[R(t)]$ as a function of round $t$. Regret is standard: $R(t) := \sum_{s \in [t]} (r^*(x_s) - r_s)$, where $r^*(x) := \max_{a \in \mathcal{A}} f^*(x, a)$, best expected reward given context $x$.

**The greedy algorithm** (`Greedy`) is defined as follows. It is initialized with $T_0 \geq 1$ rounds of warm-up data, denoted $t \in [T_0]$. [6] Each such round yields a context-arm pair $(x_t, a_t) \in \mathcal{X} \times \mathcal{A}$ chosen exogenously, and reward $r_t \in \mathbb{R}$ drawn independently from the resp. reward distribution: unit-variance Gaussian with mean $f^*(x_t, a_t)$. At each round $t > T_0$, `Greedy` computes a reward function via least-squares regression (implemented via a "regression oracle", as per Section 1):

$$f_t = \operatorname{argmin}_{f \in \mathcal{F}} \sum_{s \in [t]} (f(x_s, a_s) - r_s)^2. \tag{2.1}$$

Note that there are no ties in (2.1) with probability one over the random rewards. Once reward function $f_t$ is chosen, the algorithm chooses the best arm for $f_t$ and context $x_t$, *i.e.,*

$$a_t \in \operatorname{argmax}_{a \in \mathcal{A}} f_t(x_t, a). \tag{2.2}$$

For ease of presentation, we posit that $f(x, \cdot)$ has a unique maximizer, for each feasible function $f \in \mathcal{F}$ and each context $x \in \mathcal{X}$; call such $f$ *best-arm-unique*. (Our results can be adapted to allow for reward functions with multiple best arms, see Appendix A.)

**Notation.** Let $K$ be the number of arms; identify the action set as $\mathcal{A} = [K]$. The number of times a given arm $a$ was chosen for a given context $x$ before round $t$ is denoted $N_t(x, a)$, and the corresponding average reward is $\bar{r}_t(x, a)$. Average reward over the warm-up stage is denoted $\bar{r}_{\mathtt{warm}}(x, a) := \bar{r}_t(x, a)$ with $t = T_0 + 1$. We'll work with an alternative loss function,

$$\mathtt{MSE}_t(f) := \sum_{(x,a) \in \mathcal{X} \times \mathcal{A}} N_t(x, a) (\bar{r}_t(x, a) - f(x, a))^2. \tag{2.3}$$

Note that it is equivalent to (2.1) for minimization, in the sense that $f_t = \operatorname{argmin}_{f \in \mathcal{F}} \mathtt{MSE}_t(f)$.

## 3 Characterization for structured bandits

Let us focus on the paradigmatic special case of multi-armed bandits, call it `StructuredMAB`. Formally, there is only one context, $|\mathcal{X}| = 1$. The context can be suppressed from the notation; *e.g.,* reward functions map arms to $[0, 1]$. An arm is called *optimal* for a given reward function $f$ (or, by default, for $f = f^*$) if it maximizes expected reward $f(\cdot)$, and *suboptimal* otherwise.

We start with two key definitions. *Self-identifiability* (which drives the positive result) asserts that fixing the expected reward of any suboptimal arm identifies this arm as suboptimal.

**Definition 1** (Self-identifiability). *Fix a problem instance $(f^*, \mathcal{F})$. A suboptimal arm $a$ is called* self-identifiable *if fixing its expected reward $f^*(a)$ identifies this arm as suboptimal given $\mathcal{F}$, i.e., if arm $a$ is suboptimal for any reward function $f \in \mathcal{F}$ consistent with $f(a) = f^*(a)$. If all suboptimal arms have this property, then the problem instance is called* self-identifiable.

---

4. Whether the context distribution is known to the algorithm is inconsequential, since `Greedy` (particularly, the regression in Eq. (2.1)) does not use this knowledge. W.l.o.g., $\mathcal{X}$ is the support set of the context distribution.
5. Gaussian reward noise is a standard assumption in bandits (along with *e.g.,* 0-1 rewards), which we make for ease of presentation. Our positive results carry over to rewards with an arbitrary sub-Gaussian noise, without any modifications. Likewise, our negative results carry over to rewards $r_t \in [0, 1]$ with an arbitrary *near-uniform* distribution, *i.e.,* one specified by a p.d.f. on $[0, 1]$ which is bounded away from 0 by an absolute constant.
6. We also refer to the first $T_0$ rounds as *warm-up stage*, and the subsequent rounds as *main stage*.

A *decoy* (whose existence drives the negative result) is another reward function $f_{\text{dec}}$ such that its optimal arm $a_{\text{dec}}$ is suboptimal for $f^*$ and both reward functions coincide on this arm.

**Definition 2** (Decoy). *Let $f^*, f_{\text{dec}}$ be two reward functions, with resp. optimal arms $a^*, a_{\text{dec}}$. Call $f_{\text{dec}}$ a* decoy *for $f^*$ (with a decoy arm $a_{\text{dec}}$) if it holds that $f_{\text{dec}}(a_{\text{dec}}) = f^*(a_{\text{dec}}) < f^*(a^*)$.*

We emphasize that self-identifiability and decoys are new notions, not reducible to structural notions from prior work, see Appendix B. It is easy to see that they are equivalent, in the following sense:

**Claim 1.** *An instance $(f^*, \mathcal{F})$ is self-identifiable if and only if $f^*$ has no decoy in $\mathcal{F}$.*

In our characterization, the complexity of the problem instance $(f^*, \mathcal{F})$ enters via its *function-gap*,

$$\Gamma(f^*, \mathcal{F}) = \min_{\text{functions } f \in \mathcal{F}: \, f \neq f^*} \quad \min_{\text{arms } a: \, f(a) \neq f^*(a)} \quad |f^*(a) - f(a)|. \qquad (3.1)$$

We may also write $\Gamma(f^*) = \Gamma(f^*, \mathcal{F})$ when the function class $\mathcal{F}$ is clear from context.

**Theorem 1.** *Fix a problem instance $(f^*, \mathcal{F})$ of* `StructuredMAB`.

> *(a) If the problem instance is self-identifiable, then* `Greedy` *(with any warm-up data) satisfies $\mathbb{E}[R(t)] \leq T_0 + (K/\Gamma(f^*))^2 \cdot O(\log t)$ for each round $t \in \mathbb{N}$.*

> *(b) Suppose the warm-up data consists of one sample for each arm. Assume $f^*$ has a decoy $f_{\text{dec}} \in \mathcal{F}$, with decoy arm $a_{\text{dec}}$. Then with some probability $p_{\text{dec}} > 0$ it holds that* `Greedy` *chooses $a_{\text{dec}}$ for all rounds $t \in (T_0, \infty)$. We can lower-bound $p_{\text{dec}}$ by $e^{-O(K/\Gamma^2(f_{\text{dec}}))}$.*

**Discussion.** Thus, `Greedy` succeeds, in the sense of achieving sublinear regret for any warm-up data, if and only if the problem instance is self-identifiable. Else, `Greedy` fails for some warm-up data, incurring linear expected regret. Specifically, regret is $\mathbb{E}[R(t)] \geq (t - T_0) \cdot p_{\text{dec}} \cdot (f^*(a^*) - f^*(a_{\text{dec}}))$ for each round $t \in (T_0, \infty)$, where $a^*$ is the best arm.

The correct perspective is that `Greedy` fails on every problem instance unless self-identifiability makes it intrinsically "easy". Indeed, consider any bandit algorithm that avoids playing an arm once it is identified, with high confidence, as suboptimal and having a specific expected reward. This defines a mild yet fundamental non-degeneracy condition: a reasonable bandit algorithm should never take an action that provides neither new information (exploration) nor utility from existing information (exploitation), whether it prioritizes one or balances both. The class of algorithms satisfying this condition is broad—for instance, an algorithm may continue playing some arm $a$ indefinitely as long as the reward structure *permits* this arm to be optimal. However, under self-identifiability, any algorithm satisfying this condition achieves sublinear regret (see Appendix C for details).

The failure probability $p_{\text{dec}}$ could be quite low. When there are multiple decoys $f_{\text{dec}} \in \mathcal{F}$, we could pick one (in the analysis) which maximizes function-gap $\Gamma(f_{\text{dec}})$. We present a more efficient analysis under stronger assumptions (which also applies to infinite function classes), see Section 6.

**Proof Sketch for Theorem 1(a).** We show that a suboptimal arm $a$ cannot be chosen more than $\widetilde{O}(K/\Gamma^2(f^*))$ times throughout the main stage. Indeed, suppose $a$ is chosen this many times by some round $t > T_0$. Then $\bar{r}_t(a)$, the empirical mean reward for $a$, is within $\Gamma(f^*)/2$ of its true mean $f^*(a)$ with high probability, by a standard concentration inequality. This uniquely identifies $f^*(a)$ by definition of the function-gap, which in turn identifies $a$ as a suboptimal arm for any feasible reward function. Intuitively, this should imply that $a$ cannot be chosen again. Making this implication formal is non-trivial, requiring an additional argument invoking $\text{MSE}_t(\cdot)$, as defined in (2.3).

First, we show that $\text{MSE}_t(f^*) \leq \widetilde{O}(K)$ with high probability, using concentration. Next, we observe that any reward function $f$ with $f(a) \neq f^*(a)$ will have a larger $\text{MSE}_t(\cdot)$, and therefore cannot be chosen in round $t$. It follows that $f_t(a) = f^*(a)$. Consequently, arm $a$ is suboptimal for $f_t$ (by self-identifiability), and hence cannot be chosen in round $t$.

**Proof Sketch for Theorem 1(b).** To show that `Greedy` gets permanently trapped on the decoy arm *despite reward randomness*, we define two carefully-constructed events. The first ensures that the warm-up data causes `Greedy` to misidentify $f_{\text{dec}}$ as the true reward function for all non-decoy arms:

$$E_1 = \{ |\bar{r}_{\text{warm}}(a) - f_{\text{dec}}(a)| < \Gamma(f_{\text{dec}})/2 \quad \text{for each arm } a \neq a_{\text{dec}} \}. \qquad (3.2)$$

This concerns the single warm-up sample per non-decoy arm. The second event ensures that the empirical mean of the decoy arm $a_{\mathsf{dec}}$ remains close to $f^*(a_{\mathsf{dec}})$ for all rounds after the warm-up:

$$E_2 = \{\, \forall t > T_0, \ |\bar{r}_t(a_{\mathsf{dec}}) - f^*(a_{\mathsf{dec}})| \le \Gamma(f_{\mathsf{dec}})/2 \,\}. \tag{3.3}$$

Under $E_1 \cap E_2$, `Greedy` always chooses the decoy arm. To lower-bound $\Pr[\, E_1 \cap E_2 \,]$, note that $E_2, E_1$ are independent (as they concern, resp., $a_{\mathsf{dec}}$ and all other arms), analyze each event separately.

## 4 Characterization for structured contextual bandits (`StructuredCB`)

The ideas from Section 3 need non-trivial modifications. The naive reduction to bandits — treating each contexts-to-arms mapping as a "super-arm" in `StructuredMAB`— does *not* work because `Greedy` now observes contexts. Further, such reduction would replace the dependence on $K$ in Theorem 1 with the number of mappings, *i.e.,* $K^{|\mathcal{X}|}$, whereas we effectively replace it with $K \cdot |\mathcal{X}|$.

Some notation: mappings from contexts to arms are commonly called *policies*. Let $\Pi$ denote the set of all policies. Expected reward of policy $\pi \in \Pi$ is $f^*(\pi) := \mathbb{E}_x[\, f^*(x, \pi(x))\,]$, where the expectation is over the fixed distribution of context arrivals. A policy $\pi$ is called *optimal* for reward function $f$ if it maximizes expected reward $f(\pi)$, and *suboptimal* otherwise. Let $\pi^*$ be the optimal policy for $f^*$. Note that $\pi(x) \in \mathrm{argmax}_{a \in \mathcal{A}} f(x, \cdot)$ for each context $x$, which is unique by assumption. .

`Greedy` can be described in terms of policies: it chooses policy $\pi_t$ in each round $t$, before seeing the context $x_t$, and then chooses arm $a_t = \pi_t(x_t)$. Here $\pi_t$ is the optimal policy for the $f_t$ from Eq. (2.1).

As in Section 3, the positive and negative results are driven by, resp., self-identifiability and the existence of a suitable "decoy". Let's extend these key definitions to contextual bandits.

**Definition 3** (Self-identifiability). *Fix a problem instance $(f^*, \mathcal{F})$. A suboptimal policy $\pi \in \Pi$ is called* self-identifiable *if fixing its expected rewards $f^*(x, \pi(x))$ for all contexts $x \in \mathcal{X}$ identifies this policy as suboptimal given $\mathcal{F}$. Put differently: if this policy is suboptimal for any reward function $f \in \mathcal{F}$ such that $f(x, \pi(x)) = f^*(x, \pi(x))$ for all contexts $x$. If each suboptimal policy has this property, then the problem instance is called* self-identifiable.

**Definition 4** (Decoy). *Let $f^*, f_{\mathsf{dec}}$ be two reward functions, with resp. optimal policies $\pi^*, \pi_{\mathsf{dec}}$. Call $f_{\mathsf{dec}}$ a decoy for $f^*$ (with a decoy policy $\pi_{\mathsf{dec}}$) if it holds that $f_{\mathsf{dec}}(\pi_{\mathsf{dec}}) = f^*(\pi_{\mathsf{dec}}) < f^*(\pi^*)$ and moreover $f_{\mathsf{dec}}(x, \pi_{\mathsf{dec}}(x)) = f^*(x, \pi_{\mathsf{dec}}(x))$ for all contexts $x \in \mathcal{X}$.*

In words, the decoy and $f^*$ completely coincide on the decoy policy, which is a suboptimal policy for $f^*$. The equivalence of these definitions holds word-by-word like in Claim 1.

The notion of function-gap is extended in a natural way:

$$\Gamma(f^*, \mathcal{F}) = \min_{\text{functions } f \in \mathcal{F}:\ f \ne f^*} \ \min_{(x,a) \in \mathcal{X} \times \mathcal{A}:\ f(x,a) \ne f^*(x,a)} |f^*(x,a) - f(x,a)|. \tag{4.1}$$

Our results are also parameterized by the distribution of context arrivals, particularly by the smallest arrival probability across all contexts, denoted $p_0$. (W.l.o.g., $p_0 > 0$.)

**Theorem 2.** *Fix a problem instance $(f^*, \mathcal{F})$ of `StructuredCB`. Let $X = |\mathcal{X}|$.*

(a) *If the problem instance is self-identifiable, then `Greedy` (with any warm-up data) satisfies $\mathbb{E}[\, R(t)\,] \le T_0 + (\, |\mathcal{X}|K/\Gamma(f^*)\,)^2/p_0 \cdot O(\log t)$ for each round $t \in \mathbb{N}$.*

(b) *Suppose the warm-up data consists of one sample for each context-arm pair. Assume $f^*$ has a decoy $f_{\mathsf{dec}} \in \mathcal{F}$, with decoy policy $\pi_{\mathsf{dec}}$. Then with some probability $p_{\mathsf{dec}} > 0$, `Greedy` chooses $\pi_{\mathsf{dec}}$ in all rounds $t \in (T_0, \infty)$. We have $p_{\mathsf{dec}} \ge X^{-O(KX/\Gamma^2(f_{\mathsf{dec}}))}$.*

**Remark 1.** `Greedy` *succeeds (i.e., achieves sublinear regret for any warm-up data) if and only if the problem instance is self-identifiable. Else, `Greedy` fails for some warm-up data, with linear regret:*

$$\mathbb{E}[\, R(t)\,] \ge (t - T_0) \cdot p_{\mathsf{dec}} \cdot (f^*(\pi^*) - f^*(\pi_{\mathsf{dec}})) \text{ for each round } t \in (T_0, \infty). \tag{4.2}$$

**New Proof Ideas.** For Theorem 2(a), directly applying the proof techniques from the MAB case gives a regret bound linear in $|\Pi| = K^X$. Instead, we develop a non-trivial potential argument to achieve regret bound polynomial in $KX$. For Theorem 2(b), we give new definitions of events $E_1, E_2$ extending (3.2), (3.3) by carefully accounting for contexts, and refine the deviation analysis to remove the dependence on $|\Pi|$. Proof sketches and full proofs are in Appendix F.

# 5 Interactive decision-making with arbitrary feedback

We consider Decision-Making with Structured Observations (DMSO), a general framework for sequential decision-making with a known structure [Foster et al., 2021]. It allows for arbitrary feedback observed after each round, along with the reward.[7] The root challenge is that this feedback is usually correlated with rewards. Greedy must account for these correlations, not just compute the best fit based on rewards. Our solution is to develop a natural variant of Greedy based on maximum-likelihood estimation. The analysis then becomes much more technical compared to StructuredCB, requiring us to track changes in log-likelihood and define the "model-gap" in terms of KL-divergence.

**Preliminaries.** DMSO is defined as follows. Instead of "arms" and "contexts", we have two new primitives: a policy set $\Pi$ and observation set $\mathcal{O}$. The interaction protocol is as follows: in each round $t = 1, 2, \ldots$, the algorithm selects a policy $\pi_t \in \Pi$, receives a reward $r_t \in \mathcal{R} \subset \mathbb{R}$, and observes an observation $o_t \in \mathcal{O}$. A *model* is a mapping from $\Pi$ to a distribution over $\mathbb{R} \times \mathcal{O}$. The reward-observation pair $(r_t, o_t)$ is an independent sample from distribution $M^*(\pi_t)$, where $M^*$ is the true model. The problem structure is represented as a (known) model class $\mathcal{M}$ which contains $M^*$. We assume that $\Pi, \mathcal{M}, \mathcal{R}, \mathcal{O}$ are all finite.[8] To recap, the problem instance is a pair $(M^*, \mathcal{M})$, where $\mathcal{M}$ is known but $M^*$ is not. This completes the definition of DMSO.

StructuredMAB is a simple special case of DMSO with one possible observation. StructuredCB is subsumed by interpreting the observations $o_t$ as contexts and defining $M^*(\pi)$ accordingly, to account for the distribution of context arrivals, the reward distribution, and the reward function.[9] The observations in DMSO can also include auxiliary feedback present in various bandit models studied in prior work. To wit: rewards of "atomic actions" in combinatorial semi-bandits [*e.g.,* György et al., 2007, Chen et al., 2013], per-product sales in multi-product dynamic pricing [*e.g.,* Keskin and Zeevi, 2014, den Boer, 2014], and rewards of all "adjacent" arms in bandits with graph-based feedback [Alon et al., 2013, 2015]. Moreover, the observations can include MDP trajectories in episodic reinforcement learning [see Agarwal et al., 2020, for background]. DMSO subsumes all these scenarios, under the "realizability" assumption $M^* \in \mathcal{M}$.

We use some notation. Let $f(\pi|M)$ be the expected reward for choosing policy $\pi$ under model $M$, and $f^*(\pi) := f(\pi|M^*)$. A policy is called *optimal* (under model $M$) if it maximizes $f(\cdot|M)$, and *suboptimal* otherwise. Let $\pi^*$ be an optimal policy for $M^*$. The history $\mathcal{H}_t$ at round $t$ consists of $(\pi_s, r_s, o_s)$ tuples for all rounds $s < t$. $D \overset{\mathrm{d}}{=} D'$ denotes that distributions $D, D'$ are equal.

**Modified Greedy.** The modified greedy algorithm (GreedyMLE) uses maximum-likelihood estimation (MLE) to analyze reward-observation correlations. As before, the algorithm is initialized with $T_0 \geq 1$ rounds of warm-up data, denoted $t \in [T_0]$. Each round yields a tuple $(\pi_t, r_t, o_t) \in \Pi \times \mathbb{R} \times \mathcal{O}$, where the policy $\pi_t$ is chosen exogenously, and the $(r_t, o_t)$ pair is drawn independently from the corresponding distribution $M^*(\pi_t)$. At each round $t > T_0$, the algorithm determines

$$M_t \in \operatorname{argmax}_{M \in \mathcal{M}} \mathcal{L}(M \mid \mathcal{H}_t), \tag{5.1}$$

the model with the highest likelihood $\mathcal{L}(M|\mathcal{H}_t)$ given history $\mathcal{H}_t$ (with ties broken arbitrarily).[10] Then the algorithm chooses the optimal policy given this model: $\pi_t \in \operatorname{argmax}_{\pi \in \Pi} f(\pi|M_t)$. For simplicity, we assume that the model class $\mathcal{M}$ guarantees no ties in this argmax. Here $\mathcal{L}(M|\mathcal{H}_t)$ is an algorithm-independent notion of likelihood: the probability of seeing the reward-observation pairs in history $\mathcal{H}_t$ under model $M$, if the policies in $\mathcal{H}_t$ were chosen in the resp. rounds. In a formula,

$$\mathcal{L}(M \mid \mathcal{H}_t) := \prod_{s \in [t-1]} \operatorname{Pr}_{M(\pi_s)}(r_s, o_s). \tag{5.2}$$

W.l.o.g. we can restrict $\Pi$ to policies that are optimal for some model; in particular $|\Pi| \leq |\mathcal{M}|$.

**Our characterization.** We adapt the definitions of "self-identifiability" and "decoy" so that "two models coincide on a policy" means having the same distribution of reward-observation pairs.

**Definition 5** (Self-identifiability). *Fix a problem instance $(M^*, \mathcal{M})$. A suboptimal policy $\pi$ is called* self-identifiable *if fixing distribution $M^*(\pi)$ identifies this policy as suboptimal given $\mathcal{M}$. That is: if*

---

7. Bandit formulations with partial feedback that does *not* include the reward [known as *partial monitoring*, *e.g.,* Bartók et al., 2014, Antos et al., 2013], are outside our scope.

8. Finiteness of $\mathcal{R}, \mathcal{O}$ is for ease of presentation. We can also handle infinite $\mathcal{R}, \mathcal{O}$ if all outcome distributions $M(\pi)$ have a well-defined density, and Assumption 1 is stated in terms of these densities.

9. Here we work with discrete rewards, whereas our treatment in Sections 3 and 4 assumes Gaussian rewards.

10. As in Section 2, the regression is implemented via a "regression oracle'"; we focus on statistical guarantees.

*policy $\pi$ is suboptimal for any model $M \in \mathcal{M}$ with $M(\pi) \stackrel{d}{=} M^*(\pi)$. The problem instance is called* self-identifiable *if all suboptimal policies have this property.*

**Definition 6** (Decoy). *Let $M^*, M_{\mathrm{dec}}$ be two models, with resp. optimal policies $\pi^*, \pi_{\mathrm{dec}}$. Call $M_{\mathrm{dec}}$ a* decoy *for $M^*$ (with a decoy policy $\pi_{\mathrm{dec}}$) if $M_{\mathrm{dec}}(\pi_{\mathrm{dec}}) \stackrel{d}{=} M^*(\pi_{\mathrm{dec}})$ (i.e., the two models completely coincide on $\pi_{\mathrm{dec}}$) and moreover $\pi f^*(\pi_{\mathrm{dec}}) < f^*(\pi^*)$ (i.e., $\pi_{\mathrm{dec}}$ is suboptimal for $f^*$).*

**Claim 2.** *A `DMSO` instance $(M^*, \mathcal{M})$ is self-identifiable if and only if $M^*$ has no decoy in $\mathcal{M}$.*

We define *model-gap*, a modification of function gap which tracks the difference in reward-observation distributions (expressed via KL-divergence, denoted $D_{\mathrm{KL}}$). The model gap of model $M \in \mathcal{M}$ is

$$\Gamma(M, \mathcal{M}) := \min_{M' \in \mathcal{M}, \, \pi \in \Pi: \, M(\pi) \neq M'(\pi)} D_{\mathrm{KL}}\left(M(\pi), M'(\pi)\right).$$

Our characterization needs an assumption on the ratios of probability masses: [11]

**Assumption 1.** *The ratio $\mathrm{Pr}_{M(\pi)}(r, o) / \mathrm{Pr}_{M'(\pi)}(r, o)$ is upper-bounded by $B < \infty$, for any models $M, M' \in \mathcal{M}$, any policy $\pi \in \Pi$, and any outcome $(r, o) \in \mathcal{R} \times \mathcal{O}$.*

**Theorem 3.** *Fix an instance $(M^*, \mathcal{M})$ of `DMSO` with Assumption 1 and model-gap $\Gamma = \Gamma(M^*, \mathcal{M})$.*

- *(a) If the problem instance is self-identifiable, then `GreedyMLE` (with any warm-up data) satisfies $\mathbb{E}\left[R(t)\right] \leq T_0 + (|\Pi| \ln(B)/\Gamma)^2 \cdot O\left(\ln\left(|\mathcal{M}| \cdot t\right)\right)$ for each round $t \in \mathbb{N}$.*

- *(b) Suppose the warm-up data consists of $N_0 := c_0 \cdot (\ln(B)/\Gamma)^2 \log|\mathcal{M}|$ samples for each policy, for an appropriately chosen absolute constant $c_0$ (for the total of $T_0 := N_0|\Pi|$ samples). Assume $M^*$ has a decoy $M_{\mathrm{dec}} \in \mathcal{F}$, with decoy policy $\pi_{\mathrm{dec}}$. Then with some probability $p_{\mathrm{dec}} \geq B^{-O(N_0|\Pi|)} > 0$, `GreedyMLE` chooses $\pi_{\mathrm{dec}}$ in all rounds $t \in (T_0, \infty)$.*

`GreedyMLE` succeeds (*i.e.,* achieves sublinear regret for any warm-up data) if and only if the problem instance is self-identifiable. Else, it fails for some warm-up data, with linear regret like in Eq. (4.2). We also provide a more efficient lower bound on $p_{\mathrm{dec}}$ in Theorem 3(b), replacing $B$ with a term that only concerns two relevant models, $M_{\mathrm{dec}}, M^*$ (not all of $\mathcal{M}$). Letting $D_\infty$ be the Renyi divergence,

$$p_{\mathrm{dec}} \geq e^{-O(C_{\mathrm{dec}} N_0 |\Pi|)}, \text{ where } C_{\mathrm{dec}} = \max_{\pi \in \Pi} D_\infty\left(M_{\mathrm{dec}}(\pi) \,\|\, M^*(\pi)\right) \leq \log B. \tag{5.3}$$

**Proof Sketch for Theorem 3.** We consider the likelihood of a particular model $M \in \mathcal{M}$ given the history at round $t \geq 2$, $\mathcal{L}_t(M) := \mathcal{L}(M \mid \mathcal{H}_t)$. We track the per-round change in log-likelihood:

$$\Delta \ell_t(M) := \log \mathcal{L}_{t+1}(M) - \log \mathcal{L}_t(M) = \log\left(\mathrm{Pr}_{M(\pi_t)}(r_t, o_t)\right). \tag{5.4}$$

Let $\mathcal{L}_1(\cdot) = 1$, so that (5.4) is also well-defined for round $t = 1$.

We argue that the likelihood of $M^*$ grows *faster* than that of any other model $M \in \mathcal{M}$. Specifically, we focus on $\Phi_t(M) := \mathbb{E}\left[\Delta \ell_t(M^*) - \Delta \ell_t(M)\right]$. We claim that

$$(\forall t \in \mathbb{N}) \quad \text{If } M^*(\pi_t) \stackrel{d}{=} M(\pi_t) \text{ then } \Phi_t(M) = 0 \text{ else } \Phi_t(M) \geq \Gamma. \tag{5.5}$$

In more detail: if the two models completely coincide on policy $\pi_t$, then $\Delta \ell_t(M^*) = \Delta \ell_t(M)$, and otherwise we invoke the definition of the model-gap. We use (5.5) for both parts of the theorem. The proof of Eq. (5.5) is where we directly analyze regression and invoke the model-gap.

**Part (a).** Suppose `GreedyMLE` chooses some suboptimal policy $\pi_t$ in some round $t > T_0$ of the main stage. By Eq. (5.5) and self-identifiability, it follows that $\Phi_t(M_t) \geq \Gamma$. (Indeed, by (5.5) the only alternative is $M^*(\pi_t) \stackrel{d}{=} M_t(\pi_t)$, and then self-identifiability implies that policy $\pi_t$ is suboptimal for model $M_t$, contradiction.) Likewise, we obtain that $\Phi_t(M) \geq \Gamma$ for any model $M \in \mathcal{M}$ for which policy $\pi_t$ is optimal; let $\mathcal{M}_{\mathrm{opt}}(\pi)$ be the set of all models for which policy $\pi$ is optimal.

We argue that suboptimal policies $\pi \in \Pi$ cannot be chosen "too often". Indeed, fix one such policy $\pi$. Then with high probability (w.h.p.) the likelihood of any model $M \in \mathcal{M}_{\mathrm{opt}}(\pi)$ falls below that of $M^*$, so this model cannot be chosen again. So, w.h.p. this *policy* cannot be chosen again. [12]

---

11. Related (but incomparable) assumptions on mass/density ratios are common in the literature on online/offline RL, [*e.g.,* Munos and Szepesvári, 2008, Xie and Jiang, 2021, Zhan et al., 2022, Amortila et al., 2024].
12. This last step takes a union bound over the models $M \in \mathcal{M}_{\mathrm{opt}}(\pi)$, hence $\log(\mathcal{M})$ in the regret bound.

**Part (b).** We define independent events $E_1$ and $E_2$, resp., on the warm-up process and on all rounds when the decoy is chosen, so that $E_1 \cap E_2$ guarantees that `GreedyMLE` gets forever stuck on the decoy. While this high-level plan is the same as before, its implementation is far more challenging.

To side-step some technicalities, we separate out $N_0/2$ warm-up rounds in which the decoy policy $\pi_{\mathtt{dec}}$ is chosen. Specifically, w.l.o.g. we posit that $\pi_{\mathtt{dec}}$ is chosen in the last $N_0/2$ warm-up rounds, and let $\mathcal{H}_{\mathtt{warm}} = \mathcal{H}_{T_0'+1}$, $T_0' := T_0 - N_0/2$ be the history of the rest of the warm-up.

First, we consider the "ghost process" (`ghost`) for generating $\mathcal{H}_{\mathtt{warm}}$: in each round $t \leq T_0'$, the chosen policy $\pi_t$ stays the same, but the outcome $(r_t, o_t)$ is generated according to the decoy model $M_{\mathtt{dec}}$. Under `ghost`, each round raises the likelihood $\mathcal{L}_t(M_{\mathtt{dec}})$ *more* compared to any other model $M \in \mathcal{M}$. Namely, write $\Delta\ell_t(M) = \Delta\ell_t(M \mid \mathcal{H}_t)$ explicitly as a function of history $\mathcal{H}_t$, and let

$$\Phi_t(M, M_{\mathtt{dec}}) := \mathbb{E}\left[\Delta\ell_t(M_{\mathtt{dec}} \mid \mathcal{H}_t) - \Delta\ell_t(M \mid \mathcal{H}_t)\right], \tag{5.6}$$

where $\mathcal{H}_t$ comes from `ghost`. Reusing Eq. (5.5) (with $M_{\mathtt{dec}}$ now replacing true model $M^*$), yields:

$$\text{If } M_{\mathtt{dec}}(\pi_t) \overset{\mathrm{d}}{=} M(\pi_t) \text{ then } \Phi_t(M, M_{\mathtt{dec}}) = 0 \text{ else } \Phi_t(M, M_{\mathtt{dec}}) \geq \Gamma. \tag{5.7}$$

For each model $M \in \mathcal{M}$ different from $M_{\mathtt{dec}}$, there is a policy $\pi \in \Pi$ on which these two models differ. This policy appears $N_0$ times in the warm-up data, so by Eq. (5.7) we have $\sum_{t \in [T_0']} \Phi_t(M, M_{\mathtt{dec}}) \geq \Gamma \cdot N_0$. Consequently, letting $\mathcal{M}_{\mathtt{other}} := \mathcal{M} \setminus \{M_{\mathtt{dec}}\}$, event

$$E_1 = \left\{\forall M \in \mathcal{M}_{\mathtt{other}} \quad \mathcal{L}(M_{\mathtt{dec}} \mid \mathcal{H}_{\mathtt{warm}}) > \mathcal{L}(M \mid \mathcal{H}_{\mathtt{warm}})\right\}$$

happens w.h.p. when $\mathcal{H}_{\mathtt{warm}}$ comes from `ghost`.[13] Since `ghost` and $\mathcal{H}_{\mathtt{warm}}$ have bounded Renyi divergence, we argue that with some positive probability, event $E_1$ happens under $\mathcal{H}_{\mathtt{warm}}$.

Let's analyze the rounds in which the decoy policy $\pi_{\mathtt{dec}}$ is chosen. Let $t(j)$ be the $j$-th such round, $j \in \mathbb{N}$. We'd like to argue that throughout all these rounds, the likelihood of the decoy model $M_{\mathtt{dec}}$ grows faster than that of any other model $M \in \mathcal{M}$. To this end, consider event

$$E_2 := \left\{\forall j > N_0/2, \forall M \in \mathcal{M}_{\mathtt{other}}, \quad \sum_{i \in [j]} \Psi_i(M) \geq 0\right\},$$

where $\Psi_j(M) := \Delta\ell_{t(j)}(M_{\mathtt{dec}}) - \Delta\ell_{t(j)}(M)$. Here, we restrict to $j > N_0/2$ to ensure that $E_1, E_2$ concern disjoint sets of events, and hence are independent. $E_1 \cap E_2$ implies that in each round $t > T_0$, $\mathcal{L}_t(M_{\mathtt{dec}}) > \mathcal{L}_t(M)$ for any model $M \in \mathcal{M}_{\mathtt{other}}$, and so `GreedyMLE` chooses the decoy policy.

Finally, we argue that $E_2$ happens with positive probability. W.l.o.g., the outcomes $(r_t, o_t)$ in all rounds $t = t(j)$, $j \in \mathbb{N}$ are drawn in advance from an "outcome tape".[14] We leverage Eq. (5.5) once again. Indeed, $\Psi_j(M) = 0$ for every model $M \in \mathcal{M}$ that fully coincides with $M_{\mathtt{dec}}$ on the decoy policy $\pi_{\mathtt{dec}}$, so we only need to worry about the models $M \in \mathcal{M}$ for which this is not the case. Then $\sum_{i \in [j]} \mathbb{E}[\Psi_i(M)] \geq j \cdot \Gamma$. We obtain $\sum_{i \in [j]} \mathbb{E}[\Psi_i(M)] \geq 0$ with positive-constant probability by appropriately applying concentration separately for each $j > N_0/2$ and taking a union bound.

## 6   Structured bandits with an Infinite Function Class

We obtain a partial characterization for `StructuredMAB`, which handles an arbitrary *infinite* function class $\mathcal{F}$ and yields better constants compared to Theorem 1. The success of `Greedy` requires a stronger notion of self-identifiability: *approximately* fixing the expected reward of a suboptimal arm identifies it as suboptimal. The failure of `Greedy` requires a stronger notion of a decoy function, which must lie in the "interior" of $\mathcal{F}$. The characterization is "partial" in the sense that it does not yield a full dichotomy. However, the boundary between success and failure is controlled by a tunable "margin" parameter $\varepsilon > 0$, which can be made arbitrarily small (and optimized based on the instance).

**Definition 7.** *A problem instance $(f^*, \mathcal{F})$ is $\varepsilon$-self-identifiable, $\varepsilon \geq 0$, if any suboptimal arm $a$ of $f^*$ is suboptimal for any reward function $f \in \mathcal{F}$ with $|f(a) - f^*(a)| \leq \varepsilon$. An $\varepsilon$-interior of $\mathcal{F}$, $\mathtt{int}(\mathcal{F}, \varepsilon)$ is the set of all functions $f \in \mathcal{F}$, such that any reward function $f'$ with $\|f' - f\|_2 \leq \varepsilon$ is also in $\mathcal{F}$.*

---

13. This argument invokes a concentration inequality, which in turn uses Assumption 1. Likewise, Assumption 1 is used for another application of concentration in the end of the proof sketch.
14. Its entries $j \in \mathbb{N}$ are drawn independently from $M^*(\pi_{\mathtt{dec}})$, and $(r_{t(j)}, o_{t(j)})$ is defined as the $j$-entry.

For a "continuous" function class such as linear functions or Lipschitz functions, $\mathrm{int}(\mathcal{F}, \varepsilon)$ typically includes all but an $O(\varepsilon)$-fraction of $\mathcal{F}$. The choice of the $\ell_2$ norm in the definition of $\varepsilon$-interior is not essential: any $\ell_p$ norm suffices. We provide the main theorem below; see proof in Appendix H.

**Theorem 4.** *Fix a problem instance* $(f^*, \mathcal{F})$ *of* `StructuredMAB` *with an infinite function class* $\mathcal{F}$ *(but a finite action set* $\mathcal{A}$*). For any* $\varepsilon > 0$ *(which can be optimized based on* $f^*$*):*

(a) *If the problem instance is* $\varepsilon$-*self-identifiable, then* `Greedy` *(with any warm-up data) satisfies* $\mathbb{E}[R(t)] \leq T_0 + (K/\varepsilon)^2 \cdot O(\log t)$ *for each round* $t \in \mathbb{N}$.

(b) *Suppose the warm-up data consists of one sample for each arm. Assume* $f^*$ *has a decoy* $f_{\mathrm{dec}} \in \mathrm{int}(\mathcal{F}, \varepsilon)$, *with decoy arm* $a_{\mathrm{dec}}$. *Then with some probability* $p_{\mathrm{dec}} > 0$ *it holds that* `Greedy` *chooses* $a_{\mathrm{dec}}$ *for all rounds* $t \in (T_0, \infty)$. *We can lower-bound* $p_{\mathrm{dec}}$ *by* $e^{-O(K^2/\varepsilon^2)}$.

This result mirrors Theorem 1, with the function gap replaced by $\varepsilon$, allowing for instance-dependent optimization of $\varepsilon$ and tighter bounds. The proof for part (a) carries over with simple modifications. In contrast, proving part (b) is considerably more subtle. In the infinite case, `Greedy` may not get stuck on a single reward function—it could almost surely switch among infinitely many. The key insight is that such fluctuations need not impact the arm choice: even as the predictor $f_t$ changes, the greedy selection $a_t$ may remain fixed. The proof exploits this decoupling, constructing events where the algorithm persistently selects a decoy arm, even as the greedy predictors continue to evolve.

**Discussion: challenges.** An "if and only if" characterization for arbitrary infinite function classes is very difficult. First, one can no longer rely on the function-gap being strictly positive, which is a cornerstone of our analysis in the finite case. Second, `Greedy`'s behavior can be highly unstable: the algorithm's predictor $f_t$ can fluctuate indefinitely within a continuous region of functions that are all similarly consistent with data yet induce very different greedy action choices. As a result, the intuitive logic of "getting stuck in a decoy and staying there forever" does not directly extend.

The partial characterization in Theorem 4 is our proposed route to address these challenges. The margin $\varepsilon$ serves a dual purpose: it stands in for the now-absent function-gap, and it allows us to deal with the predictor's instability (by showing that $a_t$ can remain permanently fixed).

However, this is still insufficient for a complete characterization. For many natural function classes, our framework leaves a set of instances, typically of fraction $O(\varepsilon)$, uncharacterized. The boundary between success and failure instances in a general infinite space can be highly complex; success instances can be very close to failure instances, making a sharp separation difficult. Our $\varepsilon$-interior notion is designed to provide a robust buffer around this boundary, at the cost of leaving instances within that buffer "undecided." A tight characterization in full generality would likely require a more fine-grained analysis exploiting additional structural properties of the function class $\mathcal{F}$.

# 7  Conclusions

We study `Greedy` in structured bandits and characterize its asymptotic success vs failure in terms of a simple partial-identifiability property of the problem structure. Our characterization holds for arbitrary finite structures and extends to bandits with contexts and/or auxiliary feedback. In particular, we find that `Greedy` succeeds only if the problem is intrinsically "easy" for any algorithm which satisfies a mild non-degeneracy condition. We also provide a partial characterization for `StructuredMAB` with infinite reward structures (and finite action sets).

Several examples, both positive and negative, instantiate our characterization for various well-studied reward structures. We find that failure tends to be a common case for *bandits*, whereas both failure and success are common for structured *contextual* bandits.

We identify three directions for further work. First, extend our characterization to infinite action sets and infinite function/model classes (ideally with a complete characterization, as discussed in Section 6). Second, consider *approximate* greedy algorithms, stemming either from approximate regression or from human behavorial biases. Such algorithms, representing myopic human behavior under behavioral biases, were studied in Banihashem et al. [2023], but only for *unstructured* multi-armed bandits. Third, while our "asymptotic" perspective enables a general characterization, *stronger regret guarantees* are desirable for particular reward structures (and are known for only a few).

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

# TECHNICAL APPENDICES

## Contents

# A `StructuredCB` with tie-breaking

Let us outline how to adjust our definitions and results to account for ties in Eq. (2.2). We assume that the ties are broken at random, with some minimal probability $q_0 > 0$ on every optimal arm (*i.e.,* every arm in $\mathrm{argmax}_{a \in \mathcal{A}} f_t(x_t, a)$). More formally, `Greedy` breaks ties in Eq. (2.2) according to an independent draw from some distribution $D_t$ over the optimal arms with minimal probability at least $q_0$. Subject to this assumption, the tie-breaking distributions $D_t$ can be arbitrary, both within a given round and from one round to another.

The positive results (Definition 3 and Theorem 2(a)) carry over word-by-word, both the statements and the proofs. The negative results (Definition 4 and Theorem 2(b)) change slightly. Essentially, whenever we invoke *the* optimal arm for decoy $f_{\mathtt{dec}}$, we need to change this to *all* optimal arms for $f_{\mathtt{dec}}$.

**Definition 8** (decoy)**.** *Let $f^*$ be a reward functions, with optimal policy $\pi^*$. Another reward function $f_{\mathtt{dec}}$ is called a* decoy *for $f^*$ if any optimal policy $\pi_{\mathtt{dec}}$ for $f_{\mathtt{dec}}$ satisfies $f_{\mathtt{dec}}(\pi_{\mathtt{dec}}) = f^*(\pi_{\mathtt{dec}}) < f^*(\pi^*)$ and moreover $f_{\mathtt{dec}}(x, \pi_{\mathtt{dec}}(x)) = f^*(x, \pi_{\mathtt{dec}}(x))$ for all contexts $x \in \mathcal{X}$.*

The equivalence of self-identifiability and not having a decoy holds as before, *i.e.,* the statement of Claim 1 carries over word-by-word. Moreover, it is still the case that "self-identifiability makes the problem easy": all of Appendix C carries over as written.

**Theorem 5** (negative)**.** *Fix a problem instance $(f^*, \mathcal{F})$ of `StructuredCB`. Suppose the warm-up data consists of one sample for each context-arm pair. Assume $f^*$ has a decoy $f_{\mathtt{dec}} \in \mathcal{F}$. Let $\Pi_{\mathtt{dec}}$ is the set of all policies that are optimal for $f_{\mathtt{dec}}$. Then with some probability $p_{\mathtt{dec}} > 0$, `Greedy` only chooses policies $\pi_t \in \Pi_{\mathtt{dec}}$ in all rounds $t \in (T_0, \infty)$. We have $p_{\mathtt{dec}} \geq X^{-O(KX/\Gamma^2(f_{\mathtt{dec}}))}$, where $X = |\mathcal{X}|$.*

Under these modifications, Remark 1 applies word-by-word. In particular, existence of a decoy implies linear regret, where each round $t$ with $\pi_t \in \Pi_{\mathtt{dec}}$ increases regret by $f^*(\pi^*) - f^*(\pi_{\mathtt{dec}})$.

**Proof of Theorem 5.** The proof of Theorem 2(b) mostly carries over, with the following minor modifications. Let $\mathcal{A}^*_{\mathtt{dec}}(x) = \mathrm{argmax}_{a \in \mathcal{A}} f_{\mathtt{dec}}(x)$ be the set of optimal arms for the decoy $f_{\mathtt{dec}}$ for a given context $x$. The two events $E_1$ and $E_2$ (as originally defined eq. (F.1) and eq. (F.2)) will be modified to be invoked on all decoy context-arm pairs.

$$E_1 = \{ \, |\bar{r}_{\mathtt{warm}}(x, a) - f_{\mathtt{dec}}(x, a)| < \Gamma(f_{\mathtt{dec}})/2 \quad \text{for each } x \in \mathcal{X} \text{ and arm } a \notin \mathcal{A}^*_{\mathtt{dec}}(x) \, \},$$
$$E_2 = \{ \, |\bar{r}_t(x, \pi_{\mathtt{dec}}(x)) - f^*(x, \pi_{\mathtt{dec}}(x))| < \Gamma(f_{\mathtt{dec}})/2 \quad \text{for each } x \in \mathcal{X}, a \in \mathcal{A}^*_{\mathtt{dec}}(x), \text{ and round } t > T_0 \, \}.$$

Analyzing the probability for event $E_1$ still follows from Lemma 20. Analyzing the probability for event $E_2$ follows from Lemma 21, but with the choice of $\sigma$ will be chosen as $\sigma = \Theta(\Gamma(f_{\mathtt{dec}})/\sqrt{\ln(|\mathcal{X}|K)})$, and we still have $\Pr[E_2] \geq 0.9$.

# B   Novelty of self-identifiability

We argue that self-identifiability is a novel notion. Specifically, we compare it to (i) knowing the optimal value, and (ii) Graves-Lai coefficient being 0.

First, one could ask if self-identifiability is equivalent to knowing the value of the best arm. However, the former does not imply the latter. Consider the simple example $\mathcal{F} = \{(3,1),(2,1)\}$. Both functions are self-identifiable in $\mathcal{F}$, but clearly the optimal value differs.

Second, consider the Graves-Lai coefficient [Graves and Lai, 1997, Wagenmaker and Foster, 2023]. Let us define it formally, for the sake of completeness. Consider DMSO, as defined in Section 5, with model class $\mathcal{M}$. Let

$$\Delta(\pi|M) = f(\pi_M|M) - f(\pi|M)$$

be the suboptimality gap for model $M$ and policy $\pi$, where $\pi_M$ is the optimal policy for $M$. Let $\mathcal{M}^{\texttt{alt}}$ be the set of models that disagree with $M$ on the optimal policy:

$$\mathcal{M}^{\texttt{alt}}(M) := \{M' \in \mathcal{M} | \pi_M \neq \pi_{M'}\}.$$

Now, the Graves-Lai coefficient is defined as

$$\texttt{GLC}(\mathcal{M}, M) = \inf_{\eta \in \mathbb{R}_+^{\Pi}} \left\{ \sum_{\pi \in \Pi} \eta_\pi \Delta(\pi|M) \mid \forall M' \in \mathcal{M}^{\texttt{alt}}(M) : \sum_{\pi \in \Pi} \eta_\pi D_{\mathrm{KL}}\left(M(\pi) \| M'(\pi)\right) \geq 1 \right\}.$$

Intuitively, the Graves-Lai coefficient measures the "verification" cost of verifying whether a given function $f^*$ (or a given model $M^*$ in the DMSO setting) is indeed the true model. The Graves-Lai coefficient being 0 implies that the learner can ascertain that $f^*$ or $M^*$ is indeed the true model by simply executing the set of optimal policies $\Pi(f^*)$ or $\Pi(M^*)$.

Now, one could ask if self-identifiability is equivalent to $\texttt{GLC}(\mathcal{M}, M) = 0$. We observe that this is not the case: the two notions are incomparable. For a counterexample, consider StructuredMAB with two arms and $\mathcal{F} = \{(2,1),(0.5,1)\}$. Problem instance $f^* = (0.5,1)$ is self-identifiable, since revealing the sub-optimal arm as having reward 0.5 immediately rules out $(2,1)$ as being the true model. But the GLC $> 0$, since to ascertain $(0.5,1)$ as being the true model one necessarily has to choose the 1st arm and experiment. On the other hand, one can see $f^* = (2,1)$ is not self-identifiable but has GLC $= 0$. In this example, Greedy succeeds when GLC $> 0$ (larger GLC suggests larger regret of the optimal algorithm in GLC-based theory) but fails when GLC $= 0$ (lower GLC suggests lower regret of the optimal algorithm in GLC-based theory)! Hence GLC *does not* capture the per-instance behavior of Greedy.

However, GLC has *some* connection to our machinery. Namely, if $\texttt{GLC}(\mathcal{F}, f) = 0$ for some reward function $f$, then $f$ necessarily cannot be a decoy for any other reward function $f^*$. That said, $\texttt{GLC}(\mathcal{F}, f)$ provides no information about whether $f$ itself admits a decoy. We believe that GLC precisely characterizes the asymptotic performance of the *optimal* algorithm [Graves and Lai, 1997, Wagenmaker and Foster, 2023], whereas self-identifiability precisely captures the asymptotic behavior of Greedy —a generally suboptimal algorithm.

# C   Self-identifiability makes the problem easy

Our characterization raises a natural question: does the success of Greedy under self-identifiability stem from the algorithm itself, from self-identifiability, or both? Put differently, when Greedy succeeds, does it make any non-trivial effort toward its success?

Surprisingly, our characterization provides a definitive negative answer: Greedy succeeds because self-identifiability makes the problem intrinsically "easy." We prove that whenever self-identifiability holds, *any* reasonable algorithm (satisfying a mild non-degeneracy condition defined blow) also achieves sublinear regret. This, in a sense, reveals the "triviality" of the greedy algorithm: it succeeds only when the problem is so easy that any reasonable algorithm would succeed.

To formalize this, we must clarify what we mean by "reasonable algorithms." Clearly, we need to exclude certain degenerate cases, such as static algorithms that pick a single arm forever, neither exploring nor exploiting information. We argue that a reasonable algorithm should at least *care about* information—whether through exploration, exploitation, or both. In other words, a reasonable

algorithm should never select an action that serves *neither* any exploration purpose (i.e., bringing new information) *nor* any exploitation purpose (i.e., utilizing existing information). This principle naturally leads to *information-aware* algorithms formally defined below.

We work in the setting of `StructuredCB`, and explain how to specialize it to `StructuredMAB`.

**Definition 9.** *Consider some round $t$ in* `StructuredCB`. *We say policy $\pi$ is $\delta$-identified-and-suboptimal if there exists a suitable concentration event which happens with probability $1 - \delta$, such that under the concentration event, its mean rewards $f^*(x, \pi(x))$ for each context $x$ are exactly identified given the current history, and moreover this identification reveals that the policy is suboptimal given the function class.*

For `StructuredMAB`, this definition specializes to defining $\delta$-identified-and-suboptimal arms.

**Definition 10.** *An algorithm for* `StructuredCB` *(resp.,* `StructuredMAB`*) is called $\delta$-information-aware if at each round, it does not choose any policy (resp., arm) that is $\delta$-identified-and-suboptimal.*

Let us define the concentration events: $\mathcal{E}_{\text{MAB}}$ for `StructuredMAB` and $\mathcal{E}_{\text{CB}}$ for `StructuredCB`:

$$\mathcal{E}_{\text{MAB}} := \{ \, |\bar{r}_t(a) - f^*(a)| > \beta_t \, ( \, N_t(a) \, ) \quad \forall a \in \mathcal{A}, \, t \in \mathbb{N} \, \}, \tag{C.1}$$

$$\mathcal{E}_{\text{CB}} := \{ \, |\bar{r}_t(x, a) - f^*(x, a)| < \beta_t \, ( \, N_t(x, a) \, ) \text{ and } N_t(x, a) > \Omega(N_t(\pi) \, p_0)$$
$$\text{with } a = \pi(x) \quad \forall x \in \mathcal{X}, \, \pi \in \Pi, \, t \in \mathbb{N} \, \}, \tag{C.2}$$

where $\beta_t(n) = \sqrt{\frac{2}{n} \log \left( \frac{10K \, |\mathcal{X}| \, t \cdot n^2}{3\delta} \right)}$ and $N_t(x)$ is the number of times context $x$ has been observed before round $t$. Here, $p_0$ is the smallest context arrival probability, like in Section 4. Note that $\mathcal{E}_{\text{MAB}}$ is just a specialization of $\mathcal{E}_{\text{CB}}$.

**Theorem 6.** *Consider* `StructuredCB` *with time horizon $T$. Any $1/T$-information-aware algorithm* `ALG` *achieves a sublinear regret $\mathbb{E}\,[\,R(T)\,]$ under self-identifiability.*

*Proof.* Assume $\mathcal{E}_{\text{CB}}$ holds. Fix any suboptimal policy $\pi$. We show $\pi$ can only be chosen $o(T)$ times.

By the definition of $\beta_t(\cdot)$ in the event $\mathcal{E}_{\text{CB}}$, there must exists some parameter $T' = \widetilde{\Theta}(1/\Gamma^2(f^*))(= o(T))$, such that $\beta_t(T') < \Gamma(f^*)$. Then, if the suboptimal policy $\pi$ is executed above the threshold $\Omega(T'/p_0)$, we have $N_t(x, a) > T'$, and consequently for any context $x$,

$$|\bar{r}_t(x, \pi(x)) - f^*(x, \pi(x))| < \beta(T') < \Gamma(f^*).$$

Then recall for any function $f$ and context-arm pair $(x, a)$, we have either $f(x, a) = f^*(x, a)$ or $|f(x, a) - f^*(x, a)| \geq \Gamma(f^*)$. This precisely means the policy $\pi$ becomes identified, and by self-identifiability, any information-aware algorithm will not keep choosing $\pi$. Hence, the total regret of the information-aware algorithm is at most $O(T'|\Pi|)$, which is sublinear $o(T)$. $\square$

# D  Examples

Let us instantiate our characterization for several well-studied reward structures from bandits literature. We consider linear, Lipschitz, and (one-dimensional) polynomial structures, for bandits as well as contextual bandits. All reward structures in this section are discretized to ensure finiteness, as required for our complete characterization in Sections 3 to 5. (While our partial characterization in Section 6 handles infinite reward structures, a secondary goal of this section is to illustrate how common infinite reward structures can be meaningfully discretized so that the complete finite-structure results become directly applicable.) The discretization is consistent across different reward functions, in the sense that all functions take values in the same (discrete) set $\mathcal{R}$, with $|\mathcal{R}| \ll |\mathcal{F}|$. This prevents a trivial form of self-identifiability that could arise if each reward function $f$ were discretized independently and inconsistently, resulting in some $f(a)$ values being unique and making $f$ self-identifiable solely due to the discretization strategy specific to $f$.[15]

---

15. For example, consider an instance $(f^*, \mathcal{F})$ being *not* self-identifiable, with its decoy $f_{\text{dec}} \in \mathcal{F}$ satisfying $f^*(a_{\text{dec}}) = f_{\text{dec}}(a_{\text{dec}}) = 0.5$. Now, suppose we discretize $f^*(a_{\text{dec}})$ using discretization step 0.1 and discretize $f_{\text{dec}}(a_{\text{dec}})$ using discretization step 0.2. After this modification, $f^*(a_{\text{dec}})$ and $f_{\text{dec}}(a_{\text{dec}})$ would no longer be equal, and self-identifiability could occur.

On a high level, we prove that decoys exist for "almost all" instances of all bandit structures that we consider (*i.e.,* linear, Lipschitz, polynomial, and quadratic). Therefore, the common case in all these bandit problems is that `Greedy` fails.

For contextual bandits (CB), our findings are more nuanced. Linear CB satisfy identifiability when the context set is sufficiently diverse (which is consistent with prior work), but admit decoys (as a somewhat common case) when the context set is "low-dimensional". In contrast, existence of decoys is the common case for Lipschitz CB. One interpretation is that self-identifiability requires both context diversity and a parametric reward structure which enables precise "global inferences" (*i.e.,* inferences about arms that are far away from those that have been sampled).

In what follows, we present each structure in a self-contained way, interpreting it as special case of our framework. Since our presentation focuses on best-arm-unique reward functions, our examples are focused similarly (except those for Linear CB). Throughout, let $[y, y']_\varepsilon$ be a uniform discretization of the $[y, y']$ interval with step $\varepsilon > 0$, namely: $[y, y']_\varepsilon := \{ \varepsilon \cdot n \in [y, y'] : n \in \mathbb{N} \}$. Likewise, we define $(y, y')_\varepsilon := \{ \varepsilon \cdot n \in (y, y') : n \in \mathbb{N} \}$.

## D.1 (Discretized) linear bandits

Linear bandits is a well-studied variant of bandits [Auer, 2002, Abe et al., 2003, Dani et al., 2008, Rusmevichientong and Tsitsiklis, 2010].[16] Formally, it is a special case of `StructuredMAB` defined as follows. Arms are real-valued vectors: $\mathcal{A} \subset \mathbb{R}^d$, where $d \in \mathbb{N}$ is the dimension. Reward functions are given by $f_\theta(a) = a \cdot \theta$ for all arms $a$, where $\theta \in \Theta \subset \mathbb{R}^d$. The parameter set $\Theta$ is known to the algorithm, so the function class is $\mathcal{F} = \{ f_\theta : \theta \in \Theta \}$. The true reward function is $f^* = f_{\theta^*}$ for some $\theta^* \in \Theta$. (Fixing $\Theta$, we interpret $\theta^*$ as a "problem instance".)

Linear bandits, as traditionally defined, let $\Theta$ be (continuously) infinite, *e.g.,* a unit $\ell_1$-ball, and sometimes consider an infinite (namely, convex) action set. Here, we consider a "discretized" version, whereby both $\Theta$ and $\mathcal{A}$ are finite. Specifically, $\Theta = ( [-1, 1]_\varepsilon \setminus \{0\} )^d$, *i.e.,* all parameter vectors in $[0, 1]^d$ with discretized non-zero coordinates. Action set $\mathcal{A}$ is an arbitrary finite subset of $[-1, 1]^d$ containing the hypercube $\{ -1, 1 \}^d$.[17] Note that each reward function $f_\theta, \theta \in \Theta$ has a unique best arm $a_\theta^* = \text{sign}(\theta) := ( \text{sign}(\theta_i) : i \in [d] ) \in \{ -1, 1 \}^d$.

We prove that linear bandits has a decoy for "almost all" problem instances.

**Lemma 1.** *Consider linear bandits with dimension $d \geq 2$, parameter set $\Theta = ( [-1, 1]_\varepsilon \setminus \{0\} )^d$, $\varepsilon \in (0, 1/4]$, and an arbitrary finite action set $\mathcal{A} \subset [-1, 1]^d$ containing the hypercube $\{ -1, 1 \}^d$. Consider an instance $\theta^* \in \Theta$ such that $\|\theta^*\|_1 - 2 \min_{i \in [d]} |\theta_i^*| \geq d\varepsilon$. Then $\theta^*$ has a decoy in $\Theta$.*

*Proof.* Let $j \in [d]$ be a coordinate with the smallest $|\theta_j^*|$. Choose arm $a_{\text{dec}} \in \{ -1, 1 \}^d$ with $(a_{\text{dec}})_i = \text{sign}(\theta_i^*)$ for all coordinates $i \neq j$, and flipping the sign for $i = j$, $(a_{\text{dec}})_j = -\text{sign}(\theta_j^*)$. Note that $\langle \theta^* | a_{\text{dec}} \rangle = \|\theta^*\|_1 - 2 \min_{i \in [d]} |\theta_i^*| \in [d\varepsilon, d]$.

Now, for any given $\alpha \in [d\varepsilon, d]_\varepsilon$ and any sign vector $v \in \{-1, 1\}^d$, there is $\theta \in \Theta$ such that $\|\theta\|_1 = \alpha$ and its signs are aligned as $\text{sign}(\theta) = v$. Thus, there exists $\theta_{\text{dec}} \in \Theta$ such that $\|\theta_{\text{dec}}\|_1 = \langle \theta^* | a_{\text{dec}} \rangle$ and $\text{sign}(\theta_{\text{dec}}) = a_{\text{dec}}$. Note that $a_{\text{dec}}$ is the best arm for $\theta_{\text{dec}}$. Moreover, $\langle \theta_{\text{dec}} | a_{\text{dec}} \rangle = \|\theta_{\text{dec}}\|_1 = \langle \theta^* | a_{\text{dec}} \rangle < \|\theta^*\|_1 = \langle \theta^* | a^* \rangle$. So, $\theta_{\text{dec}}$ is a decoy for $\theta^*$. $\square$

## D.2 (Discretized) linear contextual bandits

Linear contextual bandits (CB) are studied since [Li et al., 2010, Chu et al., 2011, Abbasi-Yadkori et al., 2011]. Formally, it is a special case of `StructuredCB` defined as follows. Each context is a tuple $x = ( x(a) \in \mathbb{R}^d : a \in \mathcal{A} ) \in \mathcal{X} \subset \mathbb{R}^{d \times K}$, where $d \in \mathbb{N}$ is the dimension and $\mathcal{X}$ is the context set. Reward functions are given by $f_\theta(x, a) = x(a) \cdot \theta$ for all context-arm pairs, where $\theta \in \Theta \subset \mathbb{R}^d$ and $\Theta$ is a known parameter set. While Linear CB are traditionally defined with (continuously) infinite $\Theta$ and $\mathcal{X}$, we need both to be finite.

---

16. We consider *stochastic* linear bandits. A more general model of *adversarial* linear bandits is studied since Awerbuch and Kleinberg [2008], McMahan and Blum [2004], see Bubeck and Cesa-Bianchi [2012, Chapter 5] for a survey.
17. For ease of exposition, we relax the requirement that expected rewards must lie in $[0, 1]$.

Like in linear bandits, the function class is $\mathcal{F} = \{\, f_\theta : \theta \in \Theta \,\}$. The true reward function is $f^* = f_{\theta^*}$ for some $\theta^* \in \Theta$, which we interpret as a "problem instance".

**Remark 2.** *For this subsection, we do not assume best-arm-uniqueness, and instead rely on the version of our characterization that allows ties in (2.2), see Appendix A.*

We show that self-identifiability holds when the context set is sufficiently diverse. Essentially, we posit that per-arm contexts $x(a)$ take values in some finite subset $S_a \subset \mathbb{R}^d$ independently across arms, and each $S_a$ spans $\mathbb{R}^d$; no further assumptions are needed.

**Lemma 2** (positive). *Consider linear CB with degree $d \geq 1$ and an arbitrary finite parameter set $\Theta \subset \mathbb{R}^d$. Suppose the context set is $\mathcal{X} = \prod_{a \in \mathcal{A}} S_a$, where $S_a \subset [-1,1]^d$ are finite "per-arm" context sets such that each $S_a$ spans $\mathbb{R}^d$. Then self-identifiability holds for all instances $\theta^* \in \Theta$.*

*Proof.* Fix some policy $\pi$. For a given context $x$, let $v(x) = x(\pi(x)) \in \mathbb{R}^d$ be the context vector produced by this policy. Let's construct a set $\mathcal{X}_0 \subset \mathcal{X}$ of contexts such that $v(\mathcal{X}_0) := \{\, v(x) : x \in \mathcal{X}_0 \,\}$, the corresponding set of context vectors, spans $R^d$. Add vectors to $\mathcal{X}_0$ one by one. Suppose currently $v(\mathcal{X}_0)$ does not span $R^d$. Then, for each arm $a \in \mathcal{A}$, the per-arm context set $S_a$ is not contained in $\mathrm{span}(v(\mathcal{X}_0))$; put differently, there exists a vector $v_a \in S_a \setminus \mathrm{span}(v(\mathcal{X}_0)) \in \mathbb{R}^d$. Let $x = (\, x(a) = v_a : \forall a \in \mathcal{A}\,) \in \mathcal{X}$ be the corresponding context. It follows that $v(x) \notin \mathrm{span}(v(\mathcal{X}_0))$. Thus, adding $x$ to the set $\mathcal{X}_0$ increases $\mathrm{span}(v(\mathcal{X}_0))$. Repeat this process till $v(\mathcal{X}_0)$ spans $R^d$.

Thus, fixing expected rewards of policy $\pi$ for all contexts in $\mathcal{X}_0$ gives a linear system of the form

$$v(x) \cdot \theta^* = \alpha(x) \qquad \forall x \in \mathcal{X}_0,$$

for some known numbers $\alpha(x)$ and vectors $v(x)$, $x \in \mathcal{X}_0$. Since these vectors span $\mathbb{R}^d$, this linear system completely determines $\theta^*$. $\qquad\square$

**Remark 3.** *In particular, Lemma 2 holds when the context set is a (very) small perturbation of one particular context $x$. For a concrete formulation, let $S(a) = \{\, x(a) + \varepsilon\, e_i : i \in [d] \,\}$ for each arm $a$ and any fixed $\varepsilon > 0$, where $e_i$, $i \in [d]$ is the coordinate-$i$ unit vector. This is consistent with positive results for* Greedy *in Linear CB with smoothed contexts [Kannan et al., 2018, Bastani et al., 2021, Raghavan et al., 2023], where "nature" adds variance-$\sigma^2$ Gaussian noise to each per-arm context vector. (*Greedy *achieves optimal regret rates which degrade as $\sigma$ increases,* e.g., $\mathbb{E}\,[\,R(T)\,] \leq \widetilde{O}(\sqrt{T}/\sigma)$.) *We provide a qualitative explanation for this phenomenon.*

On the other hand, decoys may exist when the context set $\mathcal{X}$ is degenerate. We consider $\mathcal{X} = \prod_{a \in \mathcal{A}} S_a$, like in Lemma 2, but now we posit that the per-arm sets $S_a$ do *not* span $\mathbb{R}^d$, even jointly. We prove the existence of a decoy under some additional conditions.

**Lemma 3** (negative). *Consider linear CB with parameter set $\Theta = [-1,1]^d_\varepsilon$, for some degree $d \geq 2$ and discretization step $\varepsilon \in (0, 1/2]$ with $1/\varepsilon \in \mathbb{N}$. Suppose the context set is $\mathcal{X} = \prod_{a \in \mathcal{A}} S_a$, where $S_a \subset [-1,1]^d$ are the "per-arm" context sets. Assume $\mathrm{span}(S_1, \ldots, S_{K-1}) \subset R^{d-1}$ and $S_K = \{(\,0, 0, \ldots, 0, 1\,)\}$. Then any instance $\theta^* \in \Theta$ with $\theta^*_d = 1$ and $\|\theta^*\|_1 < 2$ has a decoy in $\Theta$.*

*Proof.* Consider vector $\theta_{\mathsf{dec}} \in \Theta$ such that it coincides with $\theta^*$ on the first $d-1$ components, and $(\theta_{\mathsf{dec}})_d = -1$. We claim that $\theta_{\mathsf{dec}}$ is a decoy for $\theta^*$.

To prove this claim, fix context $x \in \mathcal{X}$. Let $a^*, a_{\mathsf{dec}}$ be some optimal arms for this context under $\theta^*$ and $\theta_{\mathsf{dec}}$, respectively. Then $a_{\mathsf{dec}} \in [K-1]$. (This is because the expected reward $x(a) \cdot \theta^*$ of arm $a$ is greater than -1 when $a \in [K-1]$, and exactly $-1$ when $a = K$.) Similarly, we show that $a^* = K$. It follows that $x(a_{\mathsf{dec}}) \cdot \theta_{\mathsf{dec}} = x(a_{\mathsf{dec}}) \cdot \theta^*$, since $\theta_{\mathsf{dec}}$ and $\theta^*$ coincide on the first $K-1$ coordinates, and the last coordinate of $x(a_{\mathsf{dec}})$ is 0. Moreover $x(a_{\mathsf{dec}}) \cdot \theta^* < 1 = x(a^*) \cdot \theta^*$. Putting this together, $x(a_{\mathsf{dec}}) \cdot \theta_{\mathsf{dec}} = x(a_{\mathsf{dec}}) \cdot \theta^* < x(a^*) \cdot \theta^*$, completing the proof. $\qquad\square$

### D.3 (Discretized) Lipschitz Bandits

*Lipschitz bandits* is a special case of `StructuredMAB` in which all reward functions $f \in \mathcal{F}$ satisfy Lipschitz condition, $|f(a) - f(a')| \leq \mathcal{D}(a, a')$, for any two arms $a, a' \in \mathcal{A}$ and some known metric $\mathcal{D}$ on $\mathcal{A}$. Introduced in Kleinberg et al. [2008], Bubeck et al. [2011], Lipschitz bandits have been studied extensively since then, see Slivkins [2019, Ch. 4.4] for a survey. The paradigmatic case is

*continuum-armed bandits* [Agrawal, 1995, Kleinberg, 2004, Auer et al., 2007], where one has action set $\mathcal{A} \subset [0, 1]$ and metric $\mathcal{D}(a, a') = L \cdot |a - a'|$, for some $L > 0$.

Lipschitz bandits, as traditionally defined, allow *all* reward functions that satisfy the Lipschitz condition, and hence require an infinite function class $\mathcal{F}$. To ensure finiteness, we impose a finite action set $\mathcal{A}$ and constrain the set of possible reward values to a discretized subset $\mathcal{R} = [0, 1]_\varepsilon$. We allow all Lipschitz functions $\mathcal{A} \to \mathcal{R}$. Further, we restrict the metric $\mathcal{D}$ to take values in the same range $\mathcal{R}$. We call this problem *discretized* Lipschitz bandits.

We show that "almost any" any best-arm-unique reward function has a best-arm-unique decoy.

**Lemma 4.** *Consider discretized Lipschitz bandits, with range $\mathcal{R} = [0, 1]_\varepsilon$ and metric $\mathcal{D}$. Let $\mathcal{F}$ be the set of all best-arm-unique Lipschitz reward functions $\mathcal{A} \to \mathcal{R}$. Consider a function $f \in \mathcal{F}$ such that $0 < f(a) < f(a^*)$ some arm $a$. Then $f$ has a decoy $f_{\text{dec}} \in \mathcal{F}$ (with decoy arm $a$).*

*Proof.* Define reward function $f_{\text{dec}}$ by $f_{\text{dec}}(a') = \min\left(0,\ f(a) - \mathcal{D}(a, a')\right)$ for all arms $a' \in \mathcal{A}$. So, $f_{\text{dec}}$ takes values in $\mathcal{R}$ and is Lipschitz w.r.t. $\mathcal{D}$ (since $\mathcal{D}$ satisfies triangle inequality); hence $f_{\text{dec}} \in \mathcal{F}$. Also, $f_{\text{dec}}$ has a unique best arm $a$ (since $f(a) > 0$ and the distance between any two distinct points is positive). Note that $f_{\text{dec}}(a) = f(a) < f(a^*)$, so $f_{\text{dec}}$ is a decoy. $\qquad\square$

This result extends seamlessly to *Lipschitz contextual bandits (CB)* [Lu et al., 2010, Slivkins, 2014], albeit with somewhat heavier notation. Formally, Lipschitz CB is a special case of `StructuredCB` which posits the Lipschitz condition for all context-arm pairs: for each reward function $f \in \mathcal{F}$,

$$|f(x, a) - f(x', a')| \le \mathcal{D}\left((x, a),\ (x', a')\right) \quad \forall x, x' \in \mathcal{X},\ a, a' \in \mathcal{A}, \tag{D.1}$$

where $\mathcal{D}$ is some known metric on $\mathcal{X} \times \mathcal{A}$. As traditionally defined, Lipschitz CB allow all reward functions which satisfy (D.1). We define *discretized* Lipschitz CB same way as above: we posit finite $\mathcal{X}, \mathcal{A}$, restrict the range of the reward functions and the metric to range $\mathcal{R} = [0, 1]_\varepsilon$, and allow all functions $f : \mathcal{X} \times \mathcal{A} \to \mathcal{R}$ which satisfy (D.1). Again, we show that "almost any" any best-arm-unique reward function has a best-arm-unique decoy.

**Lemma 5.** *Consider discretized Lipschitz CB, with range $\mathcal{R} = [0, 1]_\varepsilon$ and metric $\mathcal{D}$. Let $\mathcal{F}$ be the set of all best-arm-unique Lipschitz reward functions $\mathcal{X} \times \mathcal{A} \to \mathcal{R}$. Consider a best-arm-unique function $f \in \mathcal{F}$ such that for some policy $\pi$ we have $0 < f(x, \pi(x)) < f(x, \pi^*(x))$ for each context $x$. Then $f$ has a best-arm-unique decoy $f_{\text{dec}} \in \mathcal{F}$ (with decoy policy $\pi$).*

*Proof.* Define reward function $f_{\text{dec}}$ by $f_{\text{dec}}(x, a) = \min\left(0,\ f(x, \pi(x)) - \mathcal{D}\left((x, \pi(x)), (x, a)\right)\right)$ for all context-arm pairs $(x, a)$. Like in the proof of Lemma 4, we see that $f_{\text{dec}}$ takes values in $\mathcal{R}$ and is Lipschitz w.r.t. $\mathcal{D}$, hence $f_{\text{dec}} \in \mathcal{F}$. And it has a unique best arm $\pi(x)$ for each context $x$. Finally, $f_{\text{dec}}(x, \pi(x)) = f(x, \pi(x)) < f(x, \pi^*(x))$, so $f_{\text{dec}}$ is a decoy. $\qquad\square$

### D.4 (Discretized) polynomial bandits

*Polynomial bandits* [Huang et al., 2021, Zhao et al., 2023] is a bandit problem with real-valued arms and polynomial expected rewards.[18] We obtain a negative result for "almost all" instances of polynomial bandits, and a similar-but-cleaner result for the special case of "quadratic bandits".

We define polynomial bandits as a special case of `StructuredMAB` with action set $\mathcal{A} \subset \mathbb{R}$ and reward functions $f$ are degree-$p$ polynomials, for some degree $p \in \mathbb{N}$. Denote reward functions as $f = f_{\boldsymbol{\theta}}$, where $\boldsymbol{\theta} = (\theta_0, \ldots, \theta_p) \in \mathbb{R}^{p+1}$ is the parameter vector with $\theta_p \neq 0$, so that $f_{\boldsymbol{\theta}}(a) = \sum_{q=0}^{p} \theta_q \cdot a^q$ for all arms $a$. The function set is $\mathcal{F} = \{ f_{\boldsymbol{\theta}} : \boldsymbol{\theta} \in \Theta \}$, for some parameter set $\Theta$. Typically one allows continuously many actions and parameters, *i.e.,* an infinite reward structure.

We consider *discretized* polynomial bandits, with finite $\mathcal{A}$ and $\Theta$. The action space is $\mathcal{A} = [0, 1/2]_\varepsilon$, for some fixed discretization step $\varepsilon \in (0, \frac{1}{2p})$. The parameter set $\Theta$ needs to be discretized in a more complex way, in order to guarantee that the function class contains a decoy. Namely,

$$\Theta = \prod_{q=0}^{p} \left[ -1/q,\ 1/q \right]_{\delta(q)}, \text{ where } \quad \delta(q) = \varepsilon^{p+1-q}.$$

---

18. Huang et al. [2021], Zhao et al. [2023] considered a more general formulation of polynomial bandits, with multi-dimensional arms $a \in \mathbb{R}^d$. It was also one of the explicit special cases flagged in Parys and Golrezaei [2024].

We "bunch together" all polynomials with the same leading coefficient $\theta_p$. Specifically, denote $\Theta_\gamma = \{\, \boldsymbol{\theta} \in \Theta : \theta_p = \gamma \,\}$ and $\mathcal{F}_\gamma = \{\, f_{\boldsymbol{\theta}} : \boldsymbol{\theta} \in \Theta_\gamma \,\}$, for $\gamma \neq 0$.

We focus on reward functions $f_{\boldsymbol{\theta}}$ such that

$$a_{\boldsymbol{\theta}}^* = \arg\max_{a \in \mathcal{A}} f_{\boldsymbol{\theta}} \text{ is unique} \quad \text{and} \quad f_{\boldsymbol{\theta}}(a_{\boldsymbol{\theta}}^*) > \sup_{a \in (\max \mathcal{A}, \infty)_\varepsilon} f_{\boldsymbol{\theta}}(a);$$

call such $f_{\boldsymbol{\theta}}$ *well-shaped*. In words, the "best feasible arm in $\mathcal{A}$" is unique, and dominates any larger discretized arm.[19] (We do not attempt to characterize which polynomials are well-shaped.)

We prove that "almost any" well-shaped function $f_{\boldsymbol{\theta}} \in \mathcal{F}_\gamma$ has a well-shaped decoy in $\mathcal{F}_\gamma$, for any non-zero $\gamma$ in some (discretized) range.[20] Here, "almost all" is in the sense that every non-leading coefficient of $\boldsymbol{\theta}$ must be bounded away from the boundary by $5\varepsilon$, namely: $\theta_q \in \left[\, -1/q + 5\varepsilon,\ 1/q - 5\varepsilon\, \right]$ for all $q \neq p$. Let $\Theta_\gamma^{\mathrm{bdd}}$ be the set of all such parameter vectors $\boldsymbol{\theta} \in \Theta_\gamma$. Moreover, we consider $\boldsymbol{\theta}$ such that the best arm satisfies $a_{\boldsymbol{\theta}}^* > \varepsilon$.

**Lemma 6.** *Consider discretized polynomial bandits, as defined above, for some degree $p \geq 2$ and discretization step $\varepsilon \in (0, \frac{1}{2p})$. Fix some non-zero $\gamma \in \left[\, -1/p,\ 1/p\, \right]_\varepsilon$. Then any well-shaped reward function $f_{\boldsymbol{\theta}} \in \mathcal{F}_\gamma$ with $\boldsymbol{\theta} \in \Theta_\gamma^{\mathrm{bdd}}$ and $a_{\boldsymbol{\theta}}^* > \varepsilon$ has a well-shaped decoy in $\mathcal{F}_\gamma$.*

*Proof.* Fix one such function $f_{\boldsymbol{\theta}}$. Consider a function $f_{\mathrm{dec}} : \mathbb{R} \to \mathbb{R}$ defined by

$$f_{\mathrm{dec}}(a) \equiv f_{\boldsymbol{\theta}}(a + \varepsilon) - (\, f_{\boldsymbol{\theta}}(a_{\boldsymbol{\theta}}^*) - f_{\boldsymbol{\theta}}(a_{\boldsymbol{\theta}}^* - \varepsilon)\,), \quad \forall a \in \mathbb{R}. \tag{D.2}$$

In the rest of the proof we show that $f_{\mathrm{dec}}$ is a suitable decoy.

First, we observe that $f_{\mathrm{dec}} = f_{\boldsymbol{\theta}_{\mathrm{dec}}}$, where $\boldsymbol{\theta}_{\mathrm{dec}} \in \mathbb{R}^{p+1}$ is given by $(\boldsymbol{\theta}_{\mathrm{dec}})_p = \theta_p$,

$$(\boldsymbol{\theta}_{\mathrm{dec}})_q = \sum_{i=q}^{p} \theta_i \binom{i}{q} \varepsilon^{i-q}, \quad \forall q = \{\, p-1,\ \ldots,1\,\}, \text{ and}$$

$$(\boldsymbol{\theta}_{\mathrm{dec}})_0 = \sum_{i=0}^{p} \theta_i\, \varepsilon^i - (\, f_{\boldsymbol{\theta}}(a_{\boldsymbol{\theta}}^*) - f_{\boldsymbol{\theta}}(a_{\boldsymbol{\theta}}^* - \varepsilon)\,).$$

Second, we claim that $\boldsymbol{\theta}_{\mathrm{dec}} \in \Theta_\gamma$. Indeed, the above equations imply that all coefficients of $\boldsymbol{\theta}_{\mathrm{dec}}$ are suitably discretized: $(\boldsymbol{\theta}_{\mathrm{dec}})_q \in (-\infty, \infty)_{\delta(q)}$ for all $q \in \{\, 0,\ \ldots, p-1\,\}$. It remains to show that they are suitably bounded; this is where we use $\boldsymbol{\theta} \in \Theta^{\mathrm{bdd}}$. We argue this as follows:

- Since $|\theta_q| \leq 1/q$ for all $q = 0, \ldots, p$ and $\sum_{i=1}^{p} 1/(i!) < e \leq 3$, a simple calculation shows that $|(\boldsymbol{\theta}_{\mathrm{dec}})_q - \theta_q| \leq 3\varepsilon$ for each $q \in \{\, p-1,\ \ldots, 1\,\}$.

- Since $|\theta_q| \leq 1/q$ for all $q = 0, \ldots, p$ and $a \in [0, 1/2]$, a simple calculation shows that $f_{\boldsymbol{\theta}}(a)$ is 2-Lipchitz on $\mathcal{A}$, so $|f_{\boldsymbol{\theta}}(a_{\boldsymbol{\theta}}^*) - f_{\boldsymbol{\theta}}(a_{\boldsymbol{\theta}}^* - \varepsilon)| \leq 2\varepsilon$, and moreover $|(\boldsymbol{\theta}_{\mathrm{dec}})_0 - \theta_0| \leq 3\varepsilon + 2\varepsilon = 5\varepsilon$.

Claim proved.

Third, we prove that $f_{\mathrm{dec}}$ is well-shaped and is a decoy for $f_{\boldsymbol{\theta}}$. Indeed, Eq. (D.2) and $a_{\boldsymbol{\theta}}^* > \varepsilon$, combined with the well-shaped condition (1) $a_{\boldsymbol{\theta}}^* = \arg\max_{a \in \mathcal{A}} f_{\boldsymbol{\theta}}$ being unique and (2) $f_{\boldsymbol{\theta}}(a_{\boldsymbol{\theta}}^*) < \sup_{a \in (\max \mathcal{A}, \infty)_\varepsilon} f_{\boldsymbol{\theta}}(a)$, imply that (1) $a_{\mathrm{dec}}^* = a_{\boldsymbol{\theta}}^* - \varepsilon \in \mathcal{A}$ is the unique best arm under $f_{\mathrm{dec}}$, *i.e.,* $\arg\max_{a \in \mathcal{A}} f_{\mathrm{dec}}(a)$ and (2) $f_{\mathrm{dec}}(a_{\mathrm{dec}}^*) < \sup_{a \in (\max \mathcal{A}, \infty)_\varepsilon} f_{\mathrm{dec}}(a)$, which means that $f_{\mathrm{dec}}$ satisfies the well-shaped condition. Moreover, we have

$$f_{\mathrm{dec}}(a_{\mathrm{dec}}^*) = f_{\boldsymbol{\theta}}(a_{\mathrm{dec}}^*) < f_{\boldsymbol{\theta}}(a_{\boldsymbol{\theta}}^*),$$

where the equality holds by (D.2), and the inequality holds by the uniqueness of $a_{\boldsymbol{\theta}}^*$. $\qquad \square$

---

19. Being well-shaped is a mild condition. A sufficient condition is as follows: $\arg\max_{a \in (\infty, \infty)_\varepsilon} f_{\boldsymbol{\theta}}$ is unique and lies in $(0, 1/2)$. Note that even if $\arg\max_{a \in \mathbb{R}} f_{\boldsymbol{\theta}}$ is non-unique or falls outside $(0, 1/2]$, it is still possible that $f_{\boldsymbol{\theta}}$ is well-shaped, since $\arg\max_{a \in \mathbb{R}} f_{\boldsymbol{\theta}}$ is not necessarily in $(-\infty, \infty)_\varepsilon$.
20. As a corollary, if we consider the function set consisting of all "well-shaped reward functions in $\mathcal{F}_\gamma$", then "almost any" function in this function set has a decoy in the same function set.

### D.5 (Discretized) quadratic bandits

*Quadratic bandits* is a special case of polynomial bandits, as defined in Appendix D.4, with degree $p = 2$. Quadratic bandits (in a more general formulation, with multi-dimensional arms $a \in \mathbb{R}^d$) have been studied, as an explicit model, in Shamir [2013], Huang et al. [2021], Yu et al. [2023]. We obtain a similar negative guarantee as we do for polynomial bandits – "almost any" problem instance has a decoy – but in a cleaner formulation and a simpler proof.

Let's use a more concrete notation: reward functions are $f_{\gamma, \mu, c}$ with

$$f_{(\gamma, \mu, c)}(a) = \gamma(a - \mu)^2 + c,$$

where the leading coefficient $\gamma < 0$ determines the shape (curvature) of the function and the other two parameters $\mu, c \in [0, 1]$ determine the location of the unique global maximum (*i.e.*, $(\mu, c)$).

Discretization is similar, but slightly different. The action space is $\mathcal{A} = [0, 1]_\varepsilon$, for some fixed discretization step $\varepsilon \in (0, 1/2]$. The parameter space $\Theta$, *i.e.*, the set of feasible $(\gamma, \mu, c)$ tuples, is defined as $\gamma \in [-1, -0.5]_\varepsilon$, $\mu \in [0, 1]_\varepsilon$ and $c \in [0, 1]_{\varepsilon^3}$. Note that $\mu \in \mathcal{A}$, so any function $f_{(\gamma, \mu, c)}$ has a unique optimizer at $a = \mu \in \mathcal{A}$.

We focus on function space $\mathcal{F}_\gamma := \big\{ f_{(\gamma, \mu, c)} : (\gamma, \mu, c) \in \Theta \big\}$, grouping together all functions with the same leading coefficient $\gamma$. We prove that "almost any" function in $\mathcal{F}_\gamma$ has a decoy in $\mathcal{F}_\gamma$.

**Lemma 7.** *Consider discretized quadratic bandits, for some fixed discretization step $\varepsilon \in (0, 1/2]$. Fix any leading coefficient $\gamma \in [-1, -0.5]_\varepsilon$. Then for any reward function $f^* = f_{(\gamma, \mu, c)} \in \mathcal{F}_\gamma$, it has a decoy $f_{\mathrm{dec}} \in \mathcal{F}_\gamma$, as long as $\mu, c$ are bounded away from 0: $\mu \geq \varepsilon$ and $c \geq |\gamma|\varepsilon^2$.*

*Proof.* Consider reward function $f_{\mathrm{dec}} = f_{(\gamma, \mu - \varepsilon, c + \gamma \varepsilon^2)}$. Since $\mu \geq \varepsilon$ and $c \geq |\gamma|\varepsilon^2$, it follows that $f_{\mathrm{dec}} \in \mathcal{F}_\gamma$. Let us prove that $f_{\mathrm{dec}}$ is a decoy for $f^*$. Note that $\mu - \varepsilon$ is a suboptimal action for $f^*$ and is the optimal action for $f_{\mathrm{dec}}$. Finally, it is easy to check that $f^*(\mu - \varepsilon) = \gamma \varepsilon^2 + c = f_{\mathrm{dec}}(\mu - \varepsilon)$. $\square$

## E StructuredMAB characterization: Proof of Theorem 1

### E.1 StructuredMAB Success: Proof of Theorem 1(a)

Let us fix the time horizon $t$ and show the bound on the expected regret $\mathbb{E}[R(t)]$. Recall that $\bar{r}_t(a)$ as the empirical mean for arm $a$ and that $N_t(a)$ is the number of times arm $a$ pulled up to round $t$. Also recall the greedy algorithm is minimizing the following loss function each round:

$$\mathrm{MSE}_t(f) := \sum_{a \in [K]} N_t(a)(\bar{r}(a) - f(a))^2.$$

**Lemma 8.** *Define $\beta(n) = \sqrt{\frac{2}{n} \log \frac{\pi^2 K n^2}{3\delta}}$. With probability $1 - \delta$:*

$$\forall a, \tau, |\bar{r}_\tau(a) - f^*(a)| < \beta(N_\tau(a)).$$

*Proof.* This lemma is a standard Hoeffding plus union bound, this exact form has appeared in Jun et al. [2018]. $\square$

In the following we shall always assume the event in the previous lemma holds and choose $\delta = 1/t$.

**Lemma 9.** *Assume the event in Lemma 8, then we have the upper bound on $\mathrm{MSE}_\tau$ for each round $\tau \in [t]$.*

$$\mathrm{MSE}_\tau(f^*) \leq K \cdot O(\log t).$$

*Proof.* Note that under the event from the previous lemma, we have for each arm:

$$N_\tau(a)(\bar{r}_\tau(a) - f(a))^2 \leq N_\tau(a) \cdot \beta^2(N_\tau(a))$$
$$\leq O(\log t).$$

Then, summing over all arms completes the proof. $\square$

**Lemma 10.** *Assume the event in Lemma 8. The number of times any suboptimal arm is chosen cannot exceed $T'$ rounds, where $T'$ is some parameter with $T' = (K/\Gamma(f^*)^2) \cdot O(\log t)$.*

*Proof.* We prove this by contradiction. Consider any round $\tau$ during which some suboptimal arm $a$ is chosen above this threshold $T'$. The reward for arm $a$ is going to get concentrated within $O(\Gamma(f^*))$ to $f^*(a)$, in particular:

$$|\bar{r}_\tau(a) - f^*(a)| < \Gamma(f^*)/2.$$

Take any reward vector $f'$ such that $f'(a) \neq f(a)$. By the definition of class-gap, we have:

$$|f'(a) - f^*(a)| \geq \Gamma(f^*),$$

hence

$$|\bar{r}_\tau(a) - f^*(a)| \geq \Gamma(f^*)/2.$$

Then the cumulative loss

$$\text{MSE}_\tau(f') \geq T' \cdot (\Gamma(f^*)/2)^2 = K \cdot \Omega(\log t).$$

Therefore any $f'$ with $f'(a) \neq f(a)$ cannot possibly be minimizing $\text{MSE}_\tau(\cdot)$. That is to say, the reward vector $f_\tau$ minimizing $\text{MSE}_\tau(\cdot)$ must have $f_\tau(a) = f^*(a)$. Then, by self-identifiability, we precisely know that arm $a$ is also a suboptimal arm for the reward vector $f_\tau$. Hence, we obtain a contradiction, and arm $a$ cannot possibly be chosen this round. $\qquad\square$

We complete the proof of Theorem 1(a) as follows. The regret incurred during the warmup data is at most $T_0$. Fix any round $t > T_0$. After the warmup data, we know with probability $1 - 1/t$, any suboptimal arm can be pulled at most $(K/\Gamma(f^*)^2) \cdot O(\log t)$ times after the warmup data, and the regret is $(K/\Gamma(f^*)^2) \cdot O(\log t)$. With the remaining probability $1/t$, the regret is at most $O(t)$. Hence, the theorem follows.

### E.2  `StructuredMAB` **Failure: Proof of Theorem 1(b)**

Recall the two events are defined as

$$E_1 = \left\{\, |\bar{r}_{\texttt{warm}}(a) - f_{\texttt{dec}}(a)| < \Gamma(f_{\texttt{dec}})/2 \quad \text{for each arm } a \neq a_{\texttt{dec}} \,\right\}.$$

$$E_2 = \left\{\, \forall t > T_0,\ |\bar{r}_t(a_{\texttt{dec}}) - f^*(a_{\texttt{dec}})| \leq \Gamma(f_{\texttt{dec}})/2 \,\right\}.$$

**Lemma 11.** *Assume event $E_1$ and $E_2$ holds, then greedy algorithm only choose the decoy arm $a_{\texttt{dec}}$.*

*Proof.* The proof is by induction. Assume by round $t$, the algorithm have only choose the decoy arm $a_{\texttt{dec}}$. Note that assuming event $E_1$ and $E_2$ holds, for any reward vector $f \neq f_{\texttt{dec}}$, we will have

$$|\bar{r}_t(a) - f_{\texttt{dec}}(a)| \leq |\bar{r}_t(a) - f(a)|,$$

with at least one inequality strict for one arm. Hence $f_{\texttt{dec}}$ must (still) be the $\text{MSE}_t(\cdot)$ minimizer, and $a_{\texttt{dec}}$ will be chosen in the next round. $\qquad\square$

**Lemma 12.** *Event $E_1$ happens with probability at least $\left[\frac{\Gamma(f_{\texttt{dec}})}{\sqrt{2\pi\sigma^2}} \exp\left(-2/\sigma^2\right)\right]^{K-1}$.*

*Proof.* The random variable $\bar{r}_{\texttt{warm}}(a)$ is a gaussian variable with mean $f^*(a)$ and variance $\sigma^2$. It has a distribution density at $x$ with the following form

$$\frac{1}{\sqrt{2\pi\sigma^2}} \exp\left(-(x - f^*(a))^2/(2\sigma^2)\right).$$

For any $x$ in the interval $[f_{\texttt{dec}}(a) - \Gamma(f_{\texttt{dec}})/2, f_{\texttt{dec}}(a) + \Gamma(f_{\texttt{dec}})/2]$, by boundedness of mean reward, we have

$$|x - f^*(a)| \leq 2.$$

Then, the density of $x$ at any point on the interval $[f_{\text{dec}}(a) - \Gamma(f_{\text{dec}})/2, f_{\text{dec}}(a) + \Gamma(f_{\text{dec}})/2]$ is at least

$$\frac{1}{\sqrt{2\pi\sigma^2}} \exp(-2/\sigma^2).$$

Therefore, for any arm $a$, we have the following.

$$\Pr[|\bar{r}_{\text{warm}}(a) - f_{\text{dec}}(a)| \leq \Gamma(f_{\text{dec}})/2] \geq \frac{\Gamma(f_{\text{dec}})}{\sqrt{2\pi\sigma^2}} \exp(-2/\sigma^2).$$

Since the arms are independent, it follows that event $E_1$ happens with probability

$$\left[\frac{\Gamma(f_{\text{dec}})}{\sqrt{2\pi\sigma^2}} \exp(-2/\sigma^2)\right]^{K-1}. \qquad \square$$

**Lemma 13.** *For some appropriately chosen* $\sigma = \Theta(\Gamma(f_{\text{dec}}))$, *we have event* $E_2$ *happens with probability at least*

$$\Pr\{E_2\} \geq 0.9.$$

*Proof.* Denote the bad event

$$E_3 = \{\exists t > T_0, |\bar{r}_t - f_{\text{dec}}(a_{\text{dec}})| > \Gamma(f_{\text{dec}})/2\},$$

which is the complement of $E_2$. We will obtain an upper bound on $E_3$, therefore a lower bound on $E_2$. Note that event $E_2$ (and $E_3$) is only about the decoy arm $a_{\text{dec}}$, and recall that $f^*(a_{\text{dec}}) = f_{\text{dec}}(a_{\text{dec}})$.

By union bound,

$$\begin{aligned}
\Pr[E_3] &\leq \sum_{t=1}^{T} \Pr[|\bar{r}_t - f_{\text{dec}}(a_{\text{dec}})| > \Gamma(f^*)/2] \\
&\leq 2\sum_{t=1}^{T} \exp(-t\Gamma(f_{\text{dec}})^2/\sigma^2) \\
&\leq 2\exp(-\Gamma(f_{\text{dec}})^2/\sigma^2)/(1 - \exp(-\Gamma(f_{\text{dec}})^2/\sigma^2)).
\end{aligned}$$

Here, the second inequality is by a standard Hoeffding bound, and the last inequality is by noting that we are summing a geometric sequence.

Then, we can choose some suitable $\sigma$ with $\sigma = \Theta(\Gamma(f_{\text{dec}}))$ ensures $\Pr[E_3] < 0.1$ and that $\Pr[E_2] > 0.9$. $\qquad \square$

**Lemma 14.** *For some appropriately chosen* $\sigma = \Theta(\Gamma(f_{\text{dec}}))$, *we have the following lower bound:*

$$\Pr[E_1 \cap E_2] \geq \left[\Omega(\exp(-2/\sigma^2))\right]^{K-1}$$

*Proof.* Note that event $E_1$ and $E_2$ are independent, then, the probability of $E_1 \cap E_2$ can be obtained from the previous two lemmas. $\qquad \square$

Theorem 1(b) directly follows from the above lemmas.

## F  `StructuredCB` **characterization: Proof of Theorem 2**

We start with a proof sketch, and proceed with full proofs.

**Part (a).** Directly applying the proof technique from the MAB case results in a regret bound that is linear in $|\Pi| = K^X$. Instead, we apply a potential argument and achieve regret bound that is polynomial in $KX$. First, by a standard concentration inequality, we upper-bound the loss for $f^*$ as $\text{MSE}_t(f^*) \leq \widetilde{O}(KX)$ with high probability. Then, we use self-identifiability to argue that *if* in some round $t$ of the main stage some suboptimal policy $\pi$ is chosen, there must exist some context-arm pair $(x, \pi(x))$ that is "under-explored": appeared less than $\widetilde{O}(XK/\Gamma^2(f^*))$ times. This step carefully harnesses the structure of the contextual bandit problem. Finally, we introduce a well-designed potential function (see Lemma 18) that tracks the progress of learning over time. This

function increases whenever a suboptimal policy is executed on an under-explored context-action pair, allowing us to bound the total number of times any suboptimal policy is executed. A key challenge is that while the second step guarantees the existence of an "under-explored" context-arm pair, it does not ensure that the context actually appears when the associated policy is chosen. We address this using a supermartingale argument and the fact that each context arrives with probability at least $p_0$ in each round. Combining these steps, we upper-bound the expected number of times Greedy selects a suboptimal policy, and we bound the final expected regret via the regret decomposition lemma.

**Part (b).** As in the MAB case, we define event $E_1$ to ensure that the warm-up data misidentifies $f_{\mathrm{dec}}$ as the true reward function, and event $E_2$ that the empirical rewards of the decoy policy are tightly concentrated. The definitions are modified to account for contexts:

$$E_1 = \left\{ \left| \bar{r}_{\mathrm{warm}}(x,a) - f^\dagger(x,a) \right| < \Gamma(f^\dagger)/2 \quad \text{for each } x \in \mathcal{X} \text{ and arm } a \neq \pi^\dagger(x) \right\}, \quad \text{(F.1)}$$

$$E_2 = \left\{ \left| \bar{r}_t(x,\pi^\dagger(x)) - f^*(x,\pi^\dagger(x)) \right| < \Gamma(f^\dagger)/2 \quad \text{for each } x \in \mathcal{X} \text{ and round } t > T_0 \right\}. \quad \text{(F.2)}$$

A *decoy* context-arm pair $(x,a)$ is one with $a = \pi_{\mathrm{dec}}(x)$. $E_1$ concerns the single warm-up sample for each non-decoy context-arm pair. $E_2$ asserts that the empirical rewards are concentrated for all decoy context-arm pairs (and all rounds throughout the main stage). The two events are independent, as they concern non-overlapping sets of context-arm pairs. Greedy always chooses the decoy arm when $E_1, E_2$ happen. To lower-bound $\Pr[E_1 \cap E_2]$, invoke independence, analyze each event separately.

### F.1 StructuredCB **Success: Proof of Theorem 2(a)**

Recall $N_t(x,a)$ as the number of times that context $x$ appears and arm $a$ was chosen up until round $t$. Also recall the greedy algorithm is finding the function $f$ that minimize the following function each round: $\mathrm{MSE}_t(f) = \sum_{x,a} N_t(x,a)(\bar{r}_t(x,a) - f(x,a))^2$.

Let us fix any $t \in \mathbb{N}$. We will show the upper bound on the expected regret as stated in the theorem.

**Lemma 15.** *Fix any $\delta \in (0,1)$. Let $\beta(n) = \sqrt{\frac{2}{n} \log \frac{\pi^2 X K n^2}{3\delta}}$. Then with probability at least $1 - \delta$,*

$$\forall x, a, s, |\bar{r}_s(x,a) - f(x,a)| \leq \beta(N_s(x,a)).$$

*Proof.* The proof is similar to that of Lemma 8 in the previous section, which is a Hoeffding-style concentration bound with a union bound. We can simply treat each context-arm pair $(x,a)$ as an arm, and this directly yields the result. □

In the following, we shall assume the event in the previous lemma holds, and choose $\delta = 1/t$.

**Lemma 16.** *Assume the event in Lemma 15 holds. For any round $s \in [t]$, the cumulative loss at the true underlying function, $\mathrm{MSE}_s(f^*)$, can be upper bounded as $|\mathcal{X}|K \cdot O(\log t)$.*

*Proof.* We observe that

$$\begin{aligned}
\mathrm{MSE}_s(f^*) &= \sum_{x \in \mathcal{X}, a \in [K]} N_s(x,a)(\bar{r}_s(x,a) - f(x,a))^2 \\
&\leq \sum_{x,a} O(\log(|\mathcal{X}|Kt)) \\
&\leq |\mathcal{X}|K \cdot O(\log(|\mathcal{X}|Kt)). \qquad \square
\end{aligned}$$

**Lemma 17.** *Assume the event in Lemma 15 holds. Fix any round $s$. Let $T'$ be some suitably chosen parameter and $T' = |\mathcal{X}|K/\Gamma(f^*)^2 \cdot O(\log(|\mathcal{X}|Kt))$. Suppose Greedy executes some suboptimal policy $\pi$ in round $s$. Then there exists context $x$, such that $N_s(x,\pi(x)) < T'$.*

*Proof.* We prove this by contradiction. Suppose that a suboptimal policy $\pi$ is executed at round $s$, and further suppose that for all context $x$, we have $N_s(x,\pi(x)) \geq T'$.

By the previous lemma, we have $\forall x$,

$$|\bar{r}_t(x,\pi(x)) - f^*(x,\pi(x))| < \beta(T_0) < \Gamma(f^*)/2.$$

Consider any function $f$ such that:

$$\exists x : f(x; \pi(x)) \neq f^*(x; \pi(x)). \tag{F.3}$$

By the definition of the class gap,

$$|f(x, a) - f^*(x, \pi(x))| \geq \Gamma(f^*),$$

and then

$$|f(x, a) - \bar{r}_t(x, \pi(x))| \geq \Gamma(f^*)/2.$$

Then, the term $\mathrm{MSE}_s(f)$ can be lower bounded:

$$\mathrm{MSE}_s(f) \geq T' \cdot (\Gamma(f^*)/2)^2 \geq |\mathcal{X}|K \cdot O(\log(|\mathcal{X}|Kt)).$$

Hence, any function satisfying $Eq.$ $(F.3)$ cannot possibly minimize $\mathrm{MSE}_s(\cdot)$. In other words, the function minimizing the loss at this step $f_t$ must satisfy

$$f_t(x; \pi(x)) = f^*(x; \pi(x)).$$

Finally, the self-identifiability condition precisely tells us the policy $\pi$ must be suboptimal for $f_t$ and hence cannot be executed at round $t$. We obtain a contradiction, and the lemma is proven. $\square$

**Lemma 18.** *Conditional on the event in Lemma 15, the expected total number of times of suboptimal policy execution is no larger than $|\mathcal{X}|KT'/p_0 = (|\mathcal{X}|K/\Gamma(f^*))^2/p_0 \cdot O(\log t)$.*

*Proof.* Define the potential function as

$$M_s = \sum_{x,a} \min(N_s(x, a), T').$$

Consider any round $s$ that a suboptimal policy $\pi$ is executed, by the previous lemma, there exists a context arm pair $(x, \pi(x))$ such that $N_s(x, \pi(x)) < T'$. With probability at least $p_0$, such a context $x$ will arrive, and $M_s$ will increase by 1. Therefore, whenever a suboptimal policy is executed, with probability at least $p_0$, we will have $M_{t+1} = M_t + 1$.

Let us use the indicator variable $I_s$ to denote whether a suboptimal policy is executed in round $s$. Then $M_s$ forms a supermartingale:

$$\mathbb{E}[M_s|M_{s-1}] \geq M_{s-1} + p_0 I_s.$$

Since we have that deterministically $M_t < |\mathcal{X}|KT'$, we know that the total number of times of suboptimal policy execution $N_t = \sum_{s=1}^{t} I_s$ satisfies

$$\mathbb{E}[N_t] = \mathbb{E}[\sum_{s=1}^{t} I_s] \leq \mathbb{E}[\sum_{s=1}^{t} (\mathbb{E}[M_s \mid M_{s-1}] - M_{s-1})]/p_0 < |\mathcal{X}|KT'/p_0.$$

Hence, the total number of suboptimal policies pull is upper bounded as desired. $\square$

**Proof of Theorem 2(a).** The regret incurred in the warmup phase is at most $T_0$. With probability $1 - 1/t$, the number of suboptimal policy pulls can be bounded as in the lemma above. With the remaining $1/t$ probability the regret is at most $O(t)$. Finally, by the regret decomposition lemma (Lemma 4.5 in Lattimore and Szepesvári [2020]), we have

$$\mathbb{E}[R(t)] \leq T_0 + |\mathcal{X}|KT'/p_0 + O(1)$$
$$\leq T_0 + (|\mathcal{X}|K/\Gamma(f^*))^2/p_0 \cdot O(\log t)$$

### F.2  `StructuredCB` **Failure: Proof of Theorem 2(b)**

Recall the two events

$$E_1 = \left\{ \, |\bar{r}_{\texttt{warm}}(x,a) - f_{\texttt{dec}}(x,a)| < \Gamma(f_{\texttt{dec}})/2 \quad \text{for each } x \in \mathcal{X} \text{ and arm } a \neq \pi_{\texttt{dec}}(x) \, \right\},$$
$$E_2 = \left\{ \, |\bar{r}_t(x, \pi_{\texttt{dec}}(x)) - f^*(x, \pi_{\texttt{dec}}(x))| < \Gamma(f_{\texttt{dec}})/2 \quad \text{for each } x \in \mathcal{X} \text{ and round } t > T_0 \, \right\}.$$

**Lemma 19.** *Assume event $E_1$ and $E_2$ holds. Then the greedy algorithm only executes the decoy policy $\pi_{\texttt{dec}}$.*

*Proof.* We prove this by induction. Assume up until round $t$ the greedy algorithm only executes $\pi_{\texttt{dec}}$. Consider any other function $f \neq f_{\texttt{dec}}$. Then we must have

$$\forall x, a | \bar{r}_t(x,a) - f(x,a)| \geq |\bar{r}_t(x,a) - f_{\texttt{dec}}(x,a)|.$$

And the inequality is strict for at least one $(x, a)$ pair. Hence $f_{\texttt{dec}}$ is (still) the reward function minimizing `MSE` in round $t$, and the policy $\pi_{\texttt{dec}}$ will be executed. $\qquad\square$

**Lemma 20.** *Event $E_1$ happens with probability at least $\Omega(\frac{\Gamma(f_{\texttt{dec}}) \exp(-2/\sigma^2)}{\sigma})^{|\mathcal{X}|K}$.*

*Proof.* The proof is similar to the counterpart in multi-arm bandits. Note that $\bar{r}_{\texttt{warm}}(x,a)$ is gaussian distributed with variance $\sigma^2$. We can obtain a lower bound by directly examining the distribution density of a gaussian. $\qquad\square$

**Lemma 21.** *For some suitable chosen $\sigma = \Theta(\Gamma(f_{\texttt{dec}})/\sqrt{\ln(X)})$, event $E_2$ happens with probability 0.9.*

*Proof.* Similar to the proof for multi-arm bandits, define the event

$$E_3 = \{\exists t, x, |\bar{r}_t(x, \pi_{\texttt{dec}}(x)) - f_{\texttt{dec}}(x, \pi_{\texttt{dec}}(x))| \geq \Gamma(f_{\texttt{dec}})/2\}$$

which is the complement of event $E_2$. By a union bound,

$$\Pr[E_3] = |\mathcal{X}| \sum_{t=1}^{\infty} \exp(-t\Gamma^2(f_{\texttt{dec}})/\sigma^2)$$
$$\leq |\mathcal{X}| \exp(-\Gamma^2(f_{\texttt{dec}})/\sigma^2)/(1 - \exp(-\Gamma^2(f_{\texttt{dec}})/\sigma^2))$$

Choosing some suitable $\sigma = \Theta(\Gamma(f_{\texttt{dec}})/\sqrt{\ln X})$ ensures $\Pr[E_3] < 0.1$, and consequently $\Pr[E_2] > 0.9$. $\qquad\square$

**Lemma 22.** *We have the following lower bound:*

$$\Pr[E_1 \cap E_2] \geq \left[ \log(|\mathcal{X}|) \exp(-2(\log|\mathcal{X}|)^2/\Gamma(f_{\texttt{dec}}))^2 \right]^{|\mathcal{X}|K}.$$

*Proof.* Note that event $E_1$ and $E_2$ are independent, hence the lemma follows by the previous two lemmas. $\qquad\square$

Theorem 2(b) now follows from the above lemmas.

## G  `DMSO` **characterization: Proof of Theorem 3**

Recall that $\Delta\ell_t(M)$ is the change in log-likelihood for model $M$ in round $t$, as per (5.4). Note that

$$\Delta\ell_t(M) - \Delta\ell_t(M') = \log\left( \Pr_{M(\pi_t)}(r_t, o_t) / \Pr_{M'(\pi_t)}(r_t, o_t) \right) \in [-\log B, \log B].$$

The equality is by (5.4), and the inequality is by Assumption 1 (and this is how this assumption is invoked in our analysis). We use the notation $\sigma_0 = \log|B|$ in what follows.

### G.1  DMSO **Success: Proof of Theorem 3(a)**

The below lemma bounds the number of times any suboptimal policy can be executed.

**Lemma 23.** *Let $\pi^\circ$ be any suboptimal policy. Fix $\delta \in \left(0, \frac{1}{|\mathcal{M}|}\right)$. With probability at least $1 - |\mathcal{M}|\delta$, the policy $\pi^\circ$ can be executed for at most $O\left(\sigma_0^2\,\Gamma^{-2}\,\ln(1/\delta)\right)$ rounds.*

*Proof.* Let $\mathcal{M}^*(\pi^\circ)$ be the class of models whose optimal policy is $\pi^\circ$. We show that after $\pi^\circ$ has been executed for $T'$ rounds, any model in $\mathcal{M}^*(\pi^\circ)$ cannot be the MLE maximizer with probability $1 - |\mathcal{M}|\delta$. Let $Y_t(M)$ be the difference in increase in log-likelihood of $M^*$ and $M$ in the $t$-th round:

$$Y_t(M) = \Delta\ell_t(M^*) - \Delta\ell_t(M).$$

Note that $Y_t(M)$ is a random variable where randomness comes from random realizations of reward-outcome pairs. $Y_t(M)$ can exhibit two types of behaviors:

1. $Y_t(M) = 0$, corresponding to the case where $M(\pi_t) \overset{\mathrm{d}}{=} M^*(\pi_t)$ (*i.e.,* models $M$ and $M^*$ coincide under $\pi_t$)

2. $Y_t(M)$ is a random sub-gaussian variable with variance $\leq \sigma_0^2$ and that $\mathbb{E}[Y_t] \geq \Gamma$.

Consider rounds $s$ during which the policy $\pi^\circ$ is executed. Since $\pi^\circ$ is a suboptimal policy, during these rounds, we know that $Y_t(M)$ is of the second type for any $M$ in $\mathcal{M}^*(\pi^\circ)$. That is to say, it is a subgaussian random variable with variance upper bounded by $O(\sigma_0^2)$, and that further

$$\mathbb{E}[Y_t(M)] = D_{\mathrm{KL}}(M^*(\pi^\circ), \mathcal{M}^*(\pi^\circ)).$$

Since we have assumed $\pi^\circ$ is suboptimal for the true model $M^*$, we know that,

$$\mathbb{E}[Y_t(M)] \geq \Gamma.$$

Let $Z_t(M) = \sum_{\tau=1}^{t} Y_t(M)$.
$$\Pr[N_t(\pi^\circ) > T'] \leq \Pr[\exists s, \pi \in \mathcal{M}^*(\pi^\circ), \text{s.t. } Z_s(M(\pi)) \leq 0, N_s(\pi^\circ) = T']$$
$$\leq |\mathcal{M}|\delta.$$

Here in the last line we choose $T' = O\left(\sigma_0^2\,\Gamma^{-2}\,\ln(1/\delta)\right)$, completing the proof. $\qquad\square$

We complete the proof as follows. By a union bound, with probability $1 - |M||\Pi|\delta$, the total number of rounds all suboptimal policy can be chosen is upper bounded by

$$O\left(|\Pi|\,\sigma_0^2\,\Gamma^{-2}\,\ln(1/\delta)\right).$$

Choose $\delta = 1/(t|\Pi||\mathcal{M}|)$ and that $\log(1/\delta) = O(|\Pi|t)$, then with probability $1 - 1/t$, the total number of suboptimal policies executions can be upper bounded by

$$\frac{|\Pi|\sigma_0^2}{\Gamma^2} \cdot O(\log(|\Pi|t)).$$

### G.2  DMSO **Failure: Proof of Theorem 3(b)**

In the subsequent discussion, we define $Q(E)$ as the probability of some event $E$ occurring under the assumption that the data is generated by the decoy model $M_{\mathtt{dec}}$ (a hypothetical or ghost process). Similarly, we denote $P(E)$ as the probability of event $E$ occurring under the assumption that the data is generated by $M^*$ (the true process).

Recall that the two events are defined as follows.

$$E_1 = \left\{\forall M \in \mathcal{M}_{\mathtt{other}} \quad \mathcal{L}(M_{\mathtt{dec}} \mid \mathcal{H}_{\mathtt{warm}}) > \mathcal{L}(M \mid \mathcal{H}_{\mathtt{warm}})\right\}$$

and the event

$$E_2 := \left\{\forall j > N_0/2, \forall M \in \mathcal{M}_{\mathtt{other}}, \quad \sum_{i \in [j]} \Psi_i(M) \geq 0\right\}.$$

Where we defined $\Psi_j(M) := \Delta\ell_{t(j)}(M_{\mathtt{dec}}) - \Delta\ell_{t(j)}(M)$.

We first begin with the following concentration result. This result is stated in a general manner and not specific to our problem.

**Lemma 24.** *Let $X_1, X_2, \ldots$ be a sequence of random variables with $\mathbb{E}[X_i] > \Gamma$, and each is subgaussian with variance $\sigma^2$. Then there exists some $T'$ with $T' = \Theta(\sigma^2/\Gamma^2)$, such that with probability $1 - \delta$, for any $t > T' \cdot O(\log 1/\delta)$,*

$$\sum_{\tau=1}^{t} X_\tau > 0.$$

*Proof.* This lemma is by a standard concentration with union bound. We perform the following bounds

$$\Pr\left[\exists t > T', \sum_{s=1}^{t} X_s > 0\right] \leq \sum_{t=T'}^{\infty} \Pr\left[\sum_{s=1}^{t} X_s > 0\right]$$

$$\leq \sum_{t=T'}^{\infty} \exp\left(-t\sigma^2/\Gamma^2\right)$$

$$\leq \delta.$$

Here the last line is by choosing a suitable $T' = \Theta(\sigma^2/\Gamma^2)$ and noticing we are summing a geometric sequence. $\qquad\square$

The below lemmas lower bound the probability that the likelihood of $M_{\text{dec}}$ will be the unique highest after the warmup data (assuming under ghost process $Q$).

**Lemma 25.** *We have a lower bound on $E_1$ under the ghost process:*

$$Q(E_1) \geq 0.9.$$

*Proof.* Fix a model $M \neq M_{\text{dec}}$. There must exist at least one policy $\pi$ that discriminates $M$ and $M_{\text{dec}}$, in other words the distribution $M(\pi)$ and $M_{\text{dec}}(\pi)$ are different. Then, the expected change in log-likelihood of $M_{\text{dec}}$ is at least $\Gamma(M_{\text{dec}})$ greater than that of $M$ for each time a sample or policy $\pi$ is observed:

$$\Phi_t(M, M_{\text{dec}}) \geq \Gamma.$$

where we defined $\Phi_t(M, M_{\text{dec}})$ as per Eq. (5.6),

$$\Phi_t(M, M_{\text{dec}}) := \mathbb{E}\left[\Delta\ell_t(M_{\text{dec}} \mid \mathcal{H}_t) - \Delta\ell_t(M \mid \mathcal{H}_t)\right].$$

Moreover, we know that in each round, either $\Phi_t(M, M_{\text{dec}}) = 0$, or $\Phi_t(M, M_{\text{dec}})$ is a subgaussian random variable with mean greater than $\Gamma$. Further, during rounds when $\pi$ is sampled, the latter will happen. The policy $\pi$ is sampled for $N_0 = c_0 \cdot (\sigma_0/\Gamma)^2 \log(|\Pi||M|))$ times in the warmup phase. Then, by a standard concentration inequality

$$\Pr[\ell_{\text{warm}}(M_{\text{dec}}) - \ell_{\text{warm}}(M) < 0] < 0.1/|\mathcal{M}|.$$

Now, we take a union bound over all models $|\mathcal{M}|$, and we obtain a lower bound for event $E_1$. $\qquad\square$

What remains is the to show a lower bound for the event $E_2$. We do so in the below lemma.

**Lemma 26.** *Event $E_2$ happens with probability at least $0.9$.*

Note that event $E_2$ is only about when $\pi_{\text{dec}}$ is sampled. Since $M_{\text{dec}}(\pi_{\text{dec}}) = M(\pi_{\text{dec}})$, the ghost process coincides with the true process.

*Proof.* Fix some model $M$. If the distribution for $M(\pi_{\text{dec}})$ and $M_{\text{dec}}(\pi_{\text{dec}})$ were the same, then $\Psi_j$ would be $0$ for any $j$. Hence we can assume $M(\pi_{\text{dec}})$ and $M_{\text{dec}}(\pi_{\text{dec}})$ are two different distributions. Then $\Psi_j$ would be a subgaussian random variable with $\mathbb{E}[\Psi_j] > \Gamma(M_{\text{dec}})$. By the previous Lemma 24, the event $E_2$ holds specifically for model $M$ with probability at least $1 - \delta/|M|$. Then, by a union bound, event $E_2$ holds with probability $1 - \delta$. $\qquad\square$

**Lemma 27.** *If event $E_1$ and event $E_2$ holds, then only $\pi_{\text{dec}}$ is executed.*

*Proof.* The proof is by induction. Clearly after the warmup phase, the policy $\pi_{\text{dec}}$ is executed. Now suppose up until round $t$ the policy $\pi_{\text{dec}}$ is executed, by event $E_2$, the model $M_{\text{dec}}$ remains the log-likelihood maximizer, and hence $\pi_{\text{dec}}$ will still be chosen next round. □

**Proof of Theorem 3(b).** Now let $P$ be the true underlying process for which data is actually generated according to true model $M^*$. Recall $D_\infty(M_{\text{dec}}(\pi)|M^*(\pi)) \leq \log B$. Then on the warmup data consisting of $|\Pi|N_0$ samples, the density ratio of the ghost process and true process is bounded by $B^{|\Pi|N_0}$. Therefore, the probability of event $E_1$ can be bounded as follows.

$$P(E_1) \geq Q(E_1)/B^{|\Pi|N_0}.$$

And after warmup `GreedyMLE` only choose $\pi_{\text{dec}}$ by event $E_2$. Hence, the final probability lower bound of always choosing $\pi_{\text{dec}}$ after the warmup is $\Omega(B^{-|\Pi|N_0})$.

# H `StructuredMAB` with an Infinite Function Class: Proof of Theorem 4

## H.1 Success: Proof of Theorem 4(a)

The proof of Theorem 1(a) carries over with a modified version of Lemma 10. Thus, it suffices to state and prove this modified lemma.

**Lemma 28.** *Let $\mathcal{F}$ be an infinite function class. Assume the event in Lemma 8. The number of times any suboptimal arm is chosen cannot exceed $T'$ rounds, where $T'$ is some parameter with $T' = (K/\varepsilon^2) \cdot O(\log t)$.*

*Proof.* We prove this by contradiction. Consider any round $\tau$ during which some suboptimal arm $a$ is chosen above this threshold $T'$. The reward for arm $a$ is going to get concentrated within $O(\varepsilon)$ to $f^*(a)$, in particular:

$$|\bar{r}_\tau(a) - f^*(a)| < \varepsilon/2.$$

Take any reward vector $f'$ such that $|f(a') - f(a)| > \varepsilon$. Then we have

$$|\bar{r}_\tau(a) - f(a)| \geq \varepsilon/2.$$

Then the cumulative loss

$$\text{MSE}_\tau(f') \geq T' \cdot (\varepsilon/2)^2 = K \cdot \Omega(\log t).$$

Therefore any $f'$ with $|f'(a) - f(a)| \geq \varepsilon$ cannot possibly be minimizing $\text{MSE}_\tau(\cdot)$. That is to say, the reward vector $f_\tau$ minimizing $\text{MSE}_\tau(\cdot)$ must have $|f_\tau(a) - f^*(a)| < \varepsilon$. Then, by the strong notion of $\varepsilon$-self-identifiability, we precisely know that arm $a$ is also a suboptimal arm for the reward vector $f_\tau$. Hence, we obtain a contradiction, and arm $a$ cannot possibly be chosen this round. □

## H.2 Failure: Proof of Theorem 4(b)

The proof of Theorem 4(b) follows the same structure as Theorem 1(b) (see Appendix E.2), with a key modification: although we still aim to show that `Greedy` becomes permanently stuck on $a_{\text{dec}}$ with constant probability, the regression oracle may no longer return $f_{\text{dec}}$ exactly—its output may fluctuate around $f_{\text{dec}}$ due to reward noise and the continuity of $\mathcal{F}$. Our key insight is that, under suitable probabilistic events, these fluctuations do not change the greedy decision: the regression output may differ slightly from $f_{\text{dec}}$, but the resulting action remains $a_{\text{dec}}$.

Let us define the following events:

$$E_1 = \left\{ \left| \bar{r}_{\text{warm}}(a) - (f_{\text{dec}}(a) - \varepsilon/(2\sqrt{K})) \right| < \varepsilon/(4\sqrt{K}) \quad \text{for each arm } a \neq a_{\text{dec}} \right\},$$

$$E_2 = \left\{ \forall t > T_0, \ |\bar{r}_t(a_{\text{dec}}) - f^*(a_{\text{dec}})| \leq \varepsilon/(4\sqrt{K}) \right\}.$$

These events mirror (3.2) and (3.3), but with two changes: (1) we replace the confidence radius $\Gamma(f_{\text{dec}})/2$ by $\varepsilon/(4\sqrt{K})$, and (2) shift the baseline value of $f_{\text{dec}}(a)$ by $-\varepsilon/(2\sqrt{K})$ in $E_1$.

We begin with three probabilistic lemmas.

**Lemma 29.** *Event $E_1$ happens with probability at least $\left[\frac{\varepsilon}{2\sqrt{2\pi\sigma^2 K}}\exp\left(-2/\sigma^2\right)\right]^{K-1}$.*

*Proof.* The random variable $\bar{r}_{\mathtt{warm}}(a)$ is a gaussian variable with mean $f^*(a)$ and variance $\sigma^2$. It has a distribution density at $x$ with the following form

$$\frac{1}{\sqrt{2\pi\sigma^2}}\exp\left(-(x-f^*(a))^2/(2\sigma^2)\right).$$

For any $x$ in the interval $[f_{\mathtt{dec}}(a)-3\varepsilon/(4\sqrt{K}), f_{\mathtt{dec}}(a)-\varepsilon/(4\sqrt{K})]$, by boundedness of mean reward, we have

$$|x-f^*(a)|\leq 2.$$

Then, the density of $x$ at any point on the interval $[f_{\mathtt{dec}}(a)-3\varepsilon/(4\sqrt{K}), f_{\mathtt{dec}}(a)-\varepsilon/(4\sqrt{K})]$ is at least

$$\frac{1}{\sqrt{2\pi\sigma^2}}\exp\left(-2/\sigma^2\right).$$

Therefore, for any arm $a$, we have the following.

$$\Pr\left[\left|\bar{r}_{\mathtt{warm}}(a)-(f_{\mathtt{dec}}(a)-\varepsilon/(2\sqrt{K}))\right|<\varepsilon/(4\sqrt{K})\right]\geq\frac{\varepsilon}{2\sqrt{2\pi\sigma^2 K}}\exp\left(-2/\sigma^2\right).$$

Since the arms are independent, it follows that event $E_1$ happens with probability

$$\left[\frac{\varepsilon}{2\sqrt{2\pi\sigma^2 K}}\exp\left(-2/\sigma^2\right)\right]^{K-1}. \qquad \square$$

**Lemma 30.** *For some appropriately chosen $\sigma=\Theta(\varepsilon/\sqrt{K})$, we have event $E_2$ happens with probability at least*

$$\Pr\{E_2\}\geq 0.9.$$

*Proof.* Denote the bad event

$$E_3=\left\{\exists t>T_0, |\bar{r}_t-f_{\mathtt{dec}}(a_{\mathtt{dec}})|>\varepsilon/(4\sqrt{K})\right\},$$

which is the complement of $E_2$. We will obtain an upper bound on $E_3$, therefore a lower bound on $E_2$. Note that event $E_2$ (and $E_3$) is only about the decoy arm $a_{\mathtt{dec}}$, and recall that $f^*(a_{\mathtt{dec}})=f_{\mathtt{dec}}(a_{\mathtt{dec}})$ by the definition of a decoy.

By union bound,

$$\Pr[E_3]\leq\sum_{t=1}^{T}\Pr\left[|\bar{r}_t-f_{\mathtt{dec}}(a_{\mathtt{dec}})|>\varepsilon/(4\sqrt{K})\right]$$

$$\leq 2\sum_{t=1}^{T}\exp\left(-t\varepsilon^2/(4\sigma^2 K)\right)$$

$$\leq 2\frac{\exp\left(-\varepsilon^2/(4\sigma^2 K)\right)}{(1-\exp(-\varepsilon^2/(4\sigma^2 K)))}.$$

Here, the second inequality is by a standard Hoeffding bound, and the last inequality is by noting that we are summing a geometric sequence.

Then, we can choose some suitable $\sigma$ with $\sigma=\Theta(\varepsilon/\sqrt{K})$ ensures $\Pr[E_3]<0.1$ and that $\Pr[E_2]>0.9$. $\qquad \square$

**Lemma 31.** *For some appropriately chosen $\sigma=\Theta(\varepsilon/\sqrt{K})$, we have the following lower bound:*

$$\Pr[E_1\cap E_2]\geq\left[\Omega(\exp\left(-2/\sigma^2\right))\right]^{K-1}$$

*Proof.* Since event $E_1$ concerns all $a \neq a_{\mathsf{dec}}$ and event $E_2$ concerns $a_{\mathsf{dec}}$, we know that events $E_1$ and $E_2$ are independent. As a result,

$$
\begin{aligned}
\Pr[E_1 \cap E_2] &= \Pr[E_1]\Pr[E_2] \\
&\geq 0.9\Pr[E_1] \\
&= \Omega\left(\left[\frac{\varepsilon}{2\sqrt{2\pi(\varepsilon^2/K)K}}\exp(-2/\sigma^2)\right]^{K-1}\right) \\
&= \left[\Omega(\exp(-2/\sigma^2))\right]^{K-1},
\end{aligned}
$$

where we utilize the previous two lemmas. $\qquad\square$

Having obtained the previous three probabilistic lemmas, we now prove a crucial lemma which is an extension of Lemma 11 to the infinite $\mathcal{F}$ setting. At this point, the key insight introduced in Section 6 plays a central role in the proof.

**Lemma 32.** *Assume event $E_1$ and $E_2$ holds, then greedy algorithm only choose the decoy arm $a_{\mathsf{dec}}$.*

*Proof.* The proof is by induction. Assume by round $t$, the algorithm have only choose the decoy arm $a_{\mathsf{dec}}$. Note that assuming event $E_1$ and $E_2$ holds. Consider the reward function $f_t^{\mathsf{emp}}$ given by the empirical means: $f_t^{\mathsf{emp}}(a) = \bar{r}_t(a)$ for all $a \in \mathcal{A}$. By the induction assumption, $\bar{r}_t(a) = \bar{r}_{\mathsf{warm}}(a)$ for each arm $a \neq a_{\mathsf{dec}}$. Hence, by the definition of $E_1$ and $E_2$, we have

$$
\|f_t^{\mathsf{emp}} - f_{\mathsf{dec}}\|_2 \leq \sqrt{\sum_{a \in \mathcal{A}}\left(\frac{3\varepsilon}{4\sqrt{K}}\right)^2} = \frac{3\varepsilon}{4}.
$$

Since $f_{\mathsf{dec}}$ is an $\varepsilon$-interior with respect to $\mathcal{F}$, we have $f_t^{\mathsf{emp}} \in \mathcal{F}$.

Clearly, $f_t^{\mathsf{emp}} \in \mathcal{F}$ is the unique minimizer of $\mathtt{MSE}_t(\cdot)$. To see this, for any reward vector $f \neq f_t^{\mathsf{emp}}$, we will have

$$
\left|\bar{r}_t(a) - f_t^{\mathsf{emp}}\right| = 0 \leq |\bar{r}_t(a) - f(a)|,
$$

with at least one inequality strict for one arm. Hence the regression oracle will choose $f_t = f_t^{\mathsf{emp}}$.

Although $f_t^{\mathsf{emp}}$ is not the same as $f_{\mathsf{dec}}$, its optimal action is $a_{\mathsf{dec}}$ when $E_1$ and $E_2$ happen. This is because

$$
\begin{aligned}
f_t^{\mathsf{emp}}(a_{\mathsf{dec}}) &\geq f^*(a_{\mathsf{dec}}) - \frac{\varepsilon}{4\sqrt{K}} \\
&= f_{\mathsf{dec}}(a_{\mathsf{dec}}) - \frac{\varepsilon}{4\sqrt{K}} \\
&= \left(f_{\mathsf{dec}}(a_{\mathsf{dec}}) - \frac{\varepsilon}{2\sqrt{K}}\right) + \frac{\varepsilon}{4\sqrt{K}} \\
&> f_t^{\mathsf{emp}}(a)
\end{aligned}
$$

for all $a \neq a_{\mathsf{dec}}$, where the first inequality follows from the definition of $E_2$, the first equality follows from the definition of a decoy, and the last inequality follows from the definitions of $E_1$ and $f_t^{\mathsf{emp}}$. $\qquad\square$

Theorem 4(b) directly follows from the above lemmas.

