# OpenReview forum: "Greedy Algorithms for Structured Bandits: A Sharp Characterization of Asymptotic Success / Failure"
_NeurIPS.cc/2025/Conference — NeurIPS 2025 poster_

### Official Review · Reviewer_CpfC · 2025-06-30

**Clarity:** 3
**Significance:** 2
**Originality:** 3
**Rating:** 4
**Confidence:** 3

**Summary:**

This paper characterizes the problems for structured contextual bandits that reaches asymptotic success (or failure) when the greedy policy is applied. The characterization is extended to interactive decision making with arbitrary feedback and a structured bandit with infinite function class.

**Questions:**

1. Could authors highlight the challenges and newly developed techniques to extend the success and failure condition to DMSO and structured bandits with infinite function class?

2. Can the results be extended to randomized policies with exploration probability bounded by some constant?

**Ethical Concerns:**

["NO or VERY MINOR ethics concerns only"]

**Final Justification:**

The responses helped me clearly understand the novelty of the theoretical results.

**Limitations:**

Discussions on whether the self-identifiabilty is the exact and unique condition for the success of the greedy algorithm may be helpful. It seems there can be other possible conditions that determines the success or failure of the greedy algorithm.

**Paper Formatting Concerns:**

I did not find any formatting concerns in this paper.

**Quality:**

3

**Strengths And Weaknesses:**

**Strengths**

1. The results cover general cases that are not studied in previous literature.

2. The novel concepts for hardness are discovered with clear motiviations including Graves-Lai coefficent.


**Weakness**

1. Providing the challenges of extending to interactive decision making with arbitrary case and structured bandits with infinite function class may help improve the paper.

2. Applications are limited to greedy algorithm only. Although the greedy algorithm is reasonable in some cases, exploration is widely adopted for most bandit problems.

---

> ### Author Rebuttal · Authors · 2025-07-31
>
> We thank the reviewer for their positive feedback and are glad they recognized the novelty of our concepts.
>
> **On the challenges and techniques for DMSO and infinite function classes (Response to Weakness 1):**
>
> This is a great question. We are happy to highlight the core challenges and our novel techniques:
>
> For DMSO (Section 5): The root challenge is that the additional feedback may be correlated with (and hence reveals information about) rewards. Greedy must account for these correlations, not just estimated mean rewards. Our solution is to develop a non-standard variant of Greedy based on Maximum-Likelihood Estimation (MLE) (Eq. 5.1). The analysis then becomes much more technical, requiring us to track changes in log-likelihood and define the "model-gap" in terms of KL-divergence.
>
> For Infinite Function Classes (Section 6): The first challenge is that one can no longer rely on the function-gap being strictly positive, which is a cornerstone of our analysis in the finite case. The second challenge is that Greedy's behavior can be highly unstable in nature. In an infinite class $F$, the algorithm's predictor $f_t$ can fluctuate indefinitely within a continuous region of functions that are all similarly consistent with data yet induce very different greedy action choices. As a result, the intuitive logic of “getting stuck in a decoy and staying there forever” does not directly extend. Our characterization in Section 6, which introduces a "margin" $\epsilon$ via the concepts of $\epsilon$-self-identifiability and the $\epsilon$-interior decoy, is our proposed route to address these challenges. The margin $\epsilon$ serves a dual purpose: it stands in for the now-absent function-gap, and it allows us to deal with the predictor's instability. A key technical insight of our analysis is showing that even if the predictor $f_t$ never settles, the chosen greedy action can remain permanently fixed on an $\epsilon$-interior decoy arm.
>
> **On whether the results can be extended to randomized policies with exploration probability bounded by some constant (Response to Weakness 2 and Question 2):**
>
> Not really. An algorithm like epsilon-greedy, where epsilon (exploration probability) is bounded away from zero, can easily avoid the permanent traps we identify and achieve sublinear regret (in fact, for 2-armed bandits, epsilon can be as low as $\omega(1/T)$). Put differently, even a tiny probability of unrestricted exploration easily gets around the type of failure that we discuss.
>
> As we point out in the Introduction (Lines 18-30), our motivation for studying GREEDY with zero exploration is foundational (GREEDY is basic and, in some sense, desirable) as well as economic (GREEDY reflects a natural dynamics with self-interested users). By understanding when and why GREEDY fails, we can precisely identify when exploration is *necessary*. This, in turn, justifies the role of exploration in bandit algorithms and provides a foundational rationale for bandit research.
>
> That said, there's a broader open question worth spelling out: regret lower bounds for epsilon-greedy. Results/techniques from Banihashem et al (2023) seem to imply *some* lower bounds, but fall very short from matching the well-known T^{2/3} upper bound. However, this question is open even for unstructured bandits (and even for 2 arms), so it stands to reason to study it without structure.
>
>
> **On whether self-identifiability is the exact condition for success: (Response to Question 1)**
>
> Yes—for the GREEDY algorithm, we prove that self-identifiability is both a necessary and sufficient condition for success, provided that:
> (1) we define success and failure in terms of sublinear vs. linear regret, and
> (2) we consider a finite function or model class with a strictly positive function-gap or model-gap.
>
> The negation of self-identifiability is equivalent to the existence of a decoy (Claim 1), which in turn implies failure (Theorem 1b). Together, our results provide a tight "if and only if" characterization for the settings studied in Sections 3, 4, and 5. Any alternative, non-equivalent condition would contradict our theorems.
>
> As shown in Appendix D, many infinite function classes of interest admit meaningful finite analogs via discretization. Furthermore, in Section 6, we demonstrate that natural extensions of self-identifiability allow for partial characterizations of GREEDY’s success or failure in the infinite setting. For these reasons, while it is possible that other conditions govern GREEDY's behavior in certain infinite cases, we believe self-identifiability is a foundational concept.
>
> We are happy to incorporate the above discussions into the final version of our paper!

---

> > ### Comment · Reviewer_CpfC · 2025-08-07
> > **Acknowledgement of the rebuttal**
> >
> > I appreciate the authors detailed response.
> > My questions are resolved and I will keep my score as is with higher confidence.

---

### Official Review · Reviewer_Vy9h · 2025-07-03

**Clarity:** 3
**Significance:** 4
**Originality:** 4
**Rating:** 5
**Confidence:** 5

**Summary:**

This paper presents a rigorous analysis of the performance of the greedy algorithm in structured bandit problems, including standard multi-armed bandits, contextual bandits, and decision making with structured observations (DMSO). The authors introduce the novel concepts of self-identifiability and decoy arms to characterize when the greedy algorithm performs well or poorly. The analysis first focuses on a finite function class $\mathcal{F}$, then extends to the more general and realistic setting of an infinite function class by introducing the relaxed notion of $\epsilon$-self-identifiability.

The central theoretical result establishes a dichotomy:

* If the problem instance is (or is approximately) self-identifiable, the greedy algorithm achieves sublinear regret, specifically logarithmic in time.
* If a decoy arm exists, the greedy algorithm can suffer linear regret with positive probability.

**Questions:**

The greedy algorithm is known to perform well in smoothed contextual bandits. Can the notion of $\epsilon$-self-identifiability be connected to this framework?

**Ethical Concerns:**

["NO or VERY MINOR ethics concerns only"]

**Final Justification:**

I keep my rating based on the authors' response.

**Limitations:**

* The analysis of infinite function classes using $\epsilon$-self-identifiability is a promising direction, but it remains incomplete. A more thorough characterization or sufficient conditions for this relaxed notion would improve the contribution.
* There is no empirical demonstration of the proposed ideas. A simulation illustrating the impact of self-identifiability and decoys on the greedy algorithm’s performance would help validate the theory and support its practical relevance.

**Quality:**

4

**Strengths And Weaknesses:**

The theoretical framework presented in the paper is strong and well developed. The concept of self-identifiability offers a principled condition under which greedy algorithms succeed. The idea of a decoy arm is an elegant way to formalize the situations in which greedy algorithms become misled by suboptimal choices.

However, the presentation lacks clarity in several places. In particular, the definition of self-identifiability relies on the phrase “fixing its expected reward,” which may be too abstract for readers unfamiliar with function class analysis. A more intuitive explanation of what it means to fix an arm’s expected reward and how this restricts the function class would help improve accessibility. Additionally, the paper would benefit from an early, high-level explanation of why the greedy algorithm succeeds when self-identifiability holds and why it fails in the presence of a decoy.

---

> ### Author Rebuttal · Authors · 2025-07-31
>
> We are grateful to the reviewer for their positive feedback and for recognizing our theoretical framework as "strong and well developed." We will incorporate the reviewer's excellent suggestions to improve clarity in a final version (for example, by adding an intuitive example in Section 3 to concretely show how fixing an arm's expected reward restricts the function class).
>
> **1. On the connection to smoothed contextual bandits (Response to Question 1):**
>
> This is an insightful question. Prior work has shown that GREEDY performs well in smoothed linear contextual bandits (Linear CB). In Appendix D.2, we leverage **self-identifiability** to provide a principled explanation of this phenomenon and explicitly relate it to the smoothed Linear CB literature in Remark 3.
>
> More specifically, Lemma 2 in Appendix D.2 shows that self-identifiability is a typical case in Linear CB under a mild assumption on context diversity—essentially, that the feature vectors associated with arms span the space. This condition is substantially weaker than the smoothing noise assumptions required in prior work. This distinction is intuitive: whereas prior work aims to achieve near-optimal regret rates, our goal is to characterize the more fundamental dichotomy between sublinear and linear regret, for which a less restrictive condition suffices.
>
> Appendix D.2 goes further. Lemma 3 presents a case in which self-identifiability fails due to a degenerate context set, causing GREEDY to fail in Linear CB. Such scenarios are excluded by the smoothed Linear CB framework but highlight the fundamental role of self-identifiability. Moreover, Appendix D.2 describes how to discretize the Linear CB problem to fit our main theorems (which require finite function classes). The results are formally proved for the discretized setting (BTW, while we haven't stated so in the paper, variants of these lemmas hold even without discretization, and our proofs in the discretized setting require additional effort).
>
> In this sense, self-identifiability is a very valuable concept in understanding when and why GREEDY succeeds or fails in Linear CB.
>
> Naturally, since our main characterization applies formally to finite function classes, and we propose **$\epsilon$-self-identifiability** in Section 6 to address infinite classes directly, one might ask whether smoothed contexts imply not only self-identifiability but also the stronger $\epsilon$-self-identifiability in Linear CB. If true, this could lead to a fully unified theory for Linear CB without discretization. Unfortunately, significant challenges remain. Most notably, our results in Section 6 (including the definition of $\epsilon$-self-identifiability itself) are developed for non-contextual bandits, and extending them to the contextual setting remains an open problem. We are interested in pursuing this direction in future work.
>
> **2. On the limitation of the analysis for infinite function classes (Response to Limitation 1):**
>
> We agree that our characterization for infinite classes is partial, a point we discuss in the paper. As we detailed for Reviewer eWWm, a complete characterization is exceptionally challenging because, in infinite spaces, the fundamental function-gap between distinct functions can be zero, and the behavior of Greedy can be highly unstable. Our "margin"-based framework in Section 6 is a principled and non-trivial step toward such a theory, but a complete characterization would require more new ideas and tools, which we leave as a direction for future work.
>
> **3. On the lack of empirical demonstration (Response to Limitation 2):**
>
> This is a fair point. Our primary focus in this work was on establishing the theoretical foundations and proving the linear vs. sublinear regret dichotomy via a rigorous characterization. While simulations could illustrate our theory, a key challenge is how to do so faithfully without appearing cherry-picked, given the sheer generality of our results. Which particular structures should we focus on, and which problem instances within a particular structure? How to reflect the generality of our theory while keeping the simulations manageable?
>
> A useful contrast is the simulation in Banihashem et al. (2023), which is appropriately specific, focusing on unstructured 2-armed bandits—the same scope as their main theoretical results. We believe the strength of our paper lies in its theoretical generality and rigor.  We are certainly open to exploring simulations focused on specific instantiations in future work.

---

> > ### Comment · Reviewer_Vy9h · 2025-08-04
> >
> > I appreciate the authors' response. I could better understand the work based on the response.

---

### Official Review · Reviewer_eWWm · 2025-07-03

**Clarity:** 4
**Significance:** 3
**Originality:** 3
**Rating:** 5
**Confidence:** 3

**Summary:**

This paper studies the following simple yet fundamental question in the theory of bandits: under what reward structure greedy algorithm will succeed. The authors first show that for multi-armed bandits, greedy succeeds if and only if the reward structure satisfies a certain property called self-identifiability, i.e., for any suboptimal arm $a$, the reward function class implies that $a$ is suboptimal after revealing the true mean reward value of $a$. The authors further extend this result to more general settings, including contextual bandits, interactive decision-making with arbitrary feedback, and the case when the reward function class is infinite.

**Questions:**

This is more like a suggestion rather a question, but I think the authors could outline the technical challenges of obtaining a tight characterization in the infinite function class setting, together with possible routes to resolve those challenges.

**Ethical Concerns:**

["NO or VERY MINOR ethics concerns only"]

**Final Justification:**

The authors addressed my concerns, and therefore I would like to maintain my positive score.

**Limitations:**

Yes.

**Paper Formatting Concerns:**

No such concerns.

**Quality:**

3

**Strengths And Weaknesses:**

The paper is well-written, and studied a simple yet fundamental question in the theory of bandits.  In the MAB setting, the results are obtained by (i) showing greedy will succeed when the instance is self-identifiable; and (ii) greedy will get trapped on the decoy arm (whose existence would be implied by violating self-identifiability). The authors further extend the notion of self-identifiability and decoy to more general settings, including contextual bandits and interactive decision-making with arbitrary feedback, through a series of more technical involved yet intuitive definitions.

The main weakness is that the characterization is only complete for finite function class, and there is still a gap between the positive result and negative result for infinite function class. Such a limitation has been clearly discussed in the paper, and personally I do not think such weakness is significant enough to reject the paper.

---

> ### Author Rebuttal · Authors · 2025-07-31
>
> We thank the reviewer for their thoughtful and positive evaluation, and for noting that our paper studies a "simple yet fundamental question in the theory of bandits". We especially appreciate the suggestion to outline the technical challenges involved in extending our characterization to the infinite setting.
>
> **On technical challenges for a tight characterization in the infinite case (Response to Question 1):**
>
> This is an excellent suggestion. A fully tight "if and only if" characterization for arbitrary infinite function classes is very difficult. The first fundamental challenge is that one can no longer rely on the function-gap being strictly positive, which is a cornerstone of our analysis in the finite case. The second fundamental challenge is that Greedy's behavior can be highly unstable in nature. In an infinite class $F$, the algorithm's predictor $f_t$ can fluctuate indefinitely within a continuous region of functions that are all similarly consistent with data yet induce very different greedy action choices. As a result, the intuitive logic of “getting stuck in a decoy and staying there forever” does not directly extend.
>
> Our partial characterization in Section 6, which introduces a "margin" $\epsilon$ via the concepts of $\epsilon$-self-identifiability and the $\epsilon$-interior decoy, is our proposed route to address these challenges. The margin $\epsilon$ serves a dual purpose: it stands in for the now-absent function-gap, and it allows us to deal with the predictor's instability. A key technical insight of our analysis is showing that even if the predictor $f_t$ never settles, the chosen greedy action can remain permanently fixed on an ε-interior decoy arm.
>
> However, this is still insufficient for a complete characterization. For many natural function classes, our framework leaves a set of instances, typically of fraction $O(\epsilon)$, uncharacterized. The boundary between success and failure instances in a general infinite space can be highly complex; success instances can be very close to failure instances, making a sharp separation difficult. Our $\epsilon$-interior notion is designed to provide a robust buffer around this boundary, at the cost of leaving instances within that buffer "undecided."
>
> Obtaining a tight characterization in full generality would likely require a more fine-grained analysis exploiting additional structural properties of the function class $F$, which we believe is a rich direction for future work.

---

> > ### Comment · Reviewer_eWWm · 2025-08-05
> >
> > Thank you for your response. I have no further questions and decide to keep my score.

---

### Official Review · Reviewer_jdLA · 2025-07-05

**Clarity:** 3
**Significance:** 4
**Originality:** 4
**Rating:** 5
**Confidence:** 3

**Summary:**

In this work, the authors provide a characterization of the conditions under which the greedy algorithm can succeed or fail. Specifically, they propose the so-called 'self-identifiability' property as a sharp condition for when the greedy algorithm works. The settings considered include structured bandits (non-contextual MAB), contextual bandits, and contextual decision-making with general feedback.

**Questions:**

Regarding the general feedback setting, I am wondering whether the results developed in Section 5 can be extended to—or provide more concrete insights into—the graph feedback setting as studied in [1]. Specifically, given a feedback graph $G$, how do the structure of $G$ and the function class $\mathcal{F}$ jointly influence the success or failure of greedy algorithms? For instance, if $G$ is a complete graph, then the setting reduces to the full-information case, in which the greedy algorithm should succeed for general $\mathcal{F}$.



[1] Alon, Noga, et al. "Online learning with feedback graphs: Beyond bandits." Conference on Learning Theory. PMLR, 2015.

**Ethical Concerns:**

["NO or VERY MINOR ethics concerns only"]

**Final Justification:**

The questions I raised have been well addressed. I would like to keep my recommendation to accept this paper.

**Limitations:**

Yes

**Quality:**

4

**Strengths And Weaknesses:**

Strengths:

1. I believe understanding the condition of success for greedy policies is important in both theoretical and empirical aspects; thus, the paper is well motivated.


2. This work is a technically dense paper presenting a sharp and complete theory on describing the success/failure of greedy actions across different settings.


Weaknesses:

1. The main focus of the paper is the finite reward function class. While results in the general function class setting can be obtained via discretization and various examples are provided in Appendix D, it seems different function classes require different case-by-case treatments. Is it possible to have some unified framework for dealing with general function classes—even only for sufficient conditions?

---

> ### Author Rebuttal · Authors · 2025-07-30
>
> We are grateful to the reviewer for their positive assessment and for recognizing our work as a "technically dense paper presenting a sharp and complete theory." We appreciate the insightful questions, which we address below.
>
>
> **1. On a unified framework for general function classes (Response to Weakness 1):**
>
> This is a crucial point. Our main results (Theorems 1, 2, 3) deliberately focus on the finite case to establish a provably complete characterization—a sharp "if and only if" condition that provides the foundational intuition for our work. Such a complete characterization is more challenging to obtain for infinite classes.
>
> However, our work in Section 6 on infinite function classes is precisely our proposal for a unified and principled framework. The concept of **$\epsilon$-self-identifiability** (stronger than self-identifiability, see Definition 7) provides a universal sufficient conditions for the success of Greedy (Theorem 4(a)), while the absence of **$\epsilon$-interior decoy** (weaker than non-self-identifiability, see Definition 7) provides a universal necessary conditions for the success of Greedy (Theorem 4(b)). This is not a case-by-case treatment but a **single, general characterization designed for all infinite function classes** (the proofs involve new, non-trivial ideas as explained in the last paragraph of Section 6). The gap between our sufficient and necessary conditions is controlled by a tunable margin $\epsilon$. For most natural function classes, this leaves only a small (formally, an O($\epsilon$)-fraction) set of instances uncharacterized by our theory. Obtaining a complete characterization that closes this margin for *all* instances of *any* general function class remains a considerably challenging open problem for future research.
>
> Moreover, while we explain our examples in Appendix D using our main finite function class theory (via discretization, which in our opinion is a consistent and powerful method for gaining insights), all our examples for (non-contextual) bandits can also be directly explained using the extended theory in Section 6, namely Theorem 4. In particular, our conclusion "*On a high level, we prove that decoys exist for 'almost all' instances of all bandit structures that we
> consider (i.e., linear, Lipschitz, polynomial, and quadratic). Therefore, the common case in all these
> bandit problems is that Greedy fails.*" can be obtained by directly applying Theorem 4(b) to the corresponding function classes.  In these function classes, if the true function lies in the $\epsilon$-interior, then there exists a decoy that lies in the $\Omega(\epsilon)$-interior, making Greedy fail. We'll be happy to spell out these additional examples in a revision if requested.
>
> **2. On extending the DMSO results to feedback graphs (Response to Question 1):**
>
> This is an excellent question that gets to the heart of our general framework's power. Our Decision-Making with Structured Observations (DMSO) setting in Section 5 is, in fact, a strict generalization of the feedback graph model.
>
> In the feedback graph setting of Alon et al., playing an action $i$ reveals the losses of a set of other actions $N(i)$. This can be directly mapped to our DMSO framework:
>
> - A policy $\pi$ in DMSO corresponds to choosing an arm $i$ to play.
> - The observation $o_t$ in DMSO corresponds to the vector of losses for all arms in $N(i)$.
> - The model class $M$ in DMSO would encode both the function class $F$ (mapping arms to loss distributions) and the feedback graph structure $G$ (determining which losses are observed).
>
> Our theory then provides a direct answer to the reviewer's question: The success or failure of Greedy would be determined by self-identifiability within this specific DMSO instance. An instance $(M^*, M)$ is self-identifiable if, for any suboptimal policy $\pi$ (i.e., suboptimal arm $i$), observing its corresponding feedback (the losses of arms in $N(i)$) is sufficient to rule out any model in $M$ where $\pi$ is optimal.
>
> For the reviewer's specific example, if $G$ is a complete graph (full-information), then playing any arm reveals all losses. Any suboptimal arm $i$ would be easily identified as suboptimal. Therefore, any such instance is trivially self-identifiable, and our theory correctly predicts that Greedy succeeds, consistent with the full-information case.
>
> Our framework thus provides the conceptual machinery to analyze these graph structures and beyond. More concrete insights by extending our examples in Appendix D to include feedback graphs: decoy existence is the common case for Lipschitz bandits. Essentially, if there exists an arm $a$ that does not observe the true best arm $a^*$ and some vicinity thereof, then $a$ can be a decoy arm. For linear bandits, on the other hand, we get self-identifiability as long as the arms in the neighborhood $N(a)$ span the space.
>
> Thank you again for the encouraging review and sharp questions. We hope our answers further clarify the scope and significance of our contributions.

---

> > ### Comment · Reviewer_jdLA · 2025-08-01
> >
> > I thank the authors for the detailed response, and the questions I raised have been well addressed. I would like to keep my recommendation to accept this paper.

---

### Decision · Program_Chairs · 2025-09-17

**Decision:**

Accept (poster)

**Comment:**

This paper considers the contextual bandit problem with a general class of reward functions. The paper introduces a new notion of the identifiability as the general condition for the success of the greedy algorithm. While similar concerns on the limitation to finite function classes and presentation issues are shared between the reviewers, they largely agreed in the opinion that the importance of the problem and the strength of the result overtake the limitation. I expect that the authors polish the paper in the final version by carefully improving the presentation and explaining the difficulty of the current limitation, as they are well addressed in the rebuttal.